

# A simple topography-driven and calibration-free runoff generation module

Hongkai Gao[1,2,3] *, Christian Birkel [4,5], Markus Hrachowitz [6], Doerthe Tetzlaff [5], Chris Soulsby [5], Hubert H. G. Savenije[6]

[1] Key Laboratory of Geographic Information Science (Ministry of Education of China), East China Normal University, Shanghai, China.

[2] School of Geographical Sciences, East China Normal University, Shanghai, China.

[3] Julie Ann Wrigley Global Institute of Sustainability, Arizona State University PO Box 875402. Tempe, AZ 85287-5402.

[4] Department of Geography, University of Costa Rica, San José, Costa Rica

[5] Northern Rivers Institute, University of Aberdeen, Scotland.

[6] Water Resources Section, Delft University of Technology, Delft, Netherlands.

*Corresponding to Hongkai Gao (hkgao@geo.ecnu.edu.cn)

## Abstract

Reading landscapes and developing calibration-free runoff generation models that adequately reflect land surface heterogeneities remains the focus of much hydrological research. In this study, we report a novel and simple topography-driven runoff generation parameterization – the HAND-based Storage Capacity curve (HSC), that uses a topographic index (HAND, Height Above the Nearest Drainage) to identify hydrological similarity and the extent of saturated areas in catchments. The HSC can be used as a module in any conceptual rainfall-runoff model. Further, coupling the HSC parameterization with the Mass Curve Technique (MCT) to estimate root zone storage capacity ($S_{uMax}$), we developed a calibration-free runoff generation module HSC-MCT. The runoff generation modules of HBV and TOPMODEL were used for comparison purposes. The performance of these two modules (HSC and HSC-MCT) was first checked against the data-rich Bruntland Burn (BB) catchment in Scotland, which has a long time series of field-mapped saturation area extent. We found that the HSC performed better in reproducing the spatio-temporal pattern of the observed saturated areas in the BB compared to TOPMODEL. The HSC and HSC-MCT modules were subsequently tested for 323 MOPEX catchments in the US, with diverse climate, soil,



vegetation and geological characteristics. Comparing with HBV and TOPMODEL, the HSC performs better
in both calibration and validation. Despite having no calibrated parameters, the HSC-MCT module
performed comparably well with calibrated modules, highlighting the robustness of the HSC
parameterization to describe the spatial distribution of the root zone storage capacity and the efficiency
of the MCT method to estimate $S_{uMax}$. Moreover, the HSC-MCT module facilitated effective visualization
of the saturated area, which has the potential to be used for broader hydrological, ecological,
climatological, geomorphological, and biogeochemical studies.

## 1   Introduction

Determining the volume and timing of runoff generation from rainfall inputs remains a central challenge
in rainfall-runoff modelling (Beven, 2012; McDonnell, 2013). Creating a simple, calibration-free, but robust
runoff generation module has been, and continues to be, an essential pursuit of hydrological modellers.
Although we have made tremendous advances to enhance our ability on Prediction in Ungauged Basins
(PUB) (Hrachowitz et al., 2013), it is not uncommon that models become increasingly complicated in order
to capture the details of hydrological processes shown by empirical studies (McDonnell, 2007; Sivapalan,
2009). More detailed process conceptualization normally demands higher data requirements than our
standard climatological and hydrological networks can provide, leading to more calibrated parameters
and a probable increase in model uncertainty (Sivapalan, 2009).
Hydrological connectivity is a key characteristic of catchment functioning, controlling runoff generation.
It is a property emerging at larger scales, describing the temporal dynamics of how spatially
heterogeneous storage thresholds in different parts of catchments are exceeded to contribute to storm
runoff generation and how they are thus "connected to the stream" (e.g. Zehe and Blöschl, 2004;
Bracken and Croke, 2007; Lehmann et al., 2007; Zehe and Sivapalan, 2009; Ali et al., 2013; Blume and
van Meerveld, 2015). Connectivity is controlled by a multitude of factors (Ali and Roy, 2010), including
but not limited to surface (e.g. Jencso et al., 2009) and subsurface topography (e.g. Tromp-van Meerveld
and McDonnell, 2006), soils (including preferential flow networks; e.g. Zehe et al., 2006; Weiler and
McDonnell, 2007) and land cover (e.g. Imeson and Prinsen, 2004; Jencso and McGlynn, 2011; Emanuel
et al., 2014) but also by the wetness state of the system (e.g. Detty and McGuire, 2010; Penna et al.,
2011; McMillan et al., 2014; Nippgen et al., 2015).

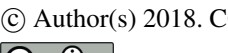


In detailed distributed hydrological bottom-up models, connectivity emerges from the interplay of
topography, soil type and water table depth. For example, TOPMODEL (Beven and Kirkby, 1979; Beven
and Freer, 2001) uses topographic information to distinguish hydrologic similarity; and SHE (Abbott et al.
1986) and tRIBS (Ivanov et al. 2004; Vivoni et al. 2005) use partial differential equations to describe the
water movement based on pressure gradients obtained by topography; and the Representative
Elementary Watershed (REW) approach divides catchment into a number of REWs to build balance and
constitutive equations for hydrological simulation (Reggiani et al., 1999; Zhang and Savenije, 2005; Tian
et al., 2008). As the relevant model parameters such as local topographic slope and hydraulic
conductivity can, in spite of several unresolved issues for example relating to the differences in the
observation and modelling scales (e.g. Beven, 1989; Zehe et al., 2014), be obtained from direct
observations, they could *in principle* be applied without calibration.
Zooming out to the macro-scale, top-down models, in contrast, are based on emergent functional
relationships that integrate system-internal heterogeneity (Sivapalan, 2005). These functional
relationships require parameters that are effective on the modelling scale and that can largely not be
directly determined with small-scale field observations (cf. Beven, 1995). Parameters in these models
are therefore traditionally determined by calibration. However, frequently the number of observed
variables for model calibration is, if available at all, limited to time series of stream flow. The absence of
more variables to constrain models results in such models being ill-posed inverse problems. Equifinality
in parameterization and in the choice of parameters then results in considerable model uncertainty (e.g.
Beven, 1993, 2006). To limit this problem and to also allow predictions in the vast majority of
catchments worldwide that remain ungauged it is therefore desirable to find ways to directly infer
effective model parameters at the modelling scale from readily available data (Hrachowitz et al., 2013).
The component that is central for establishing connectivity in most top-down models is the soil moisture
routine. Briefly, it controls the dynamics of water storage and release in the unsaturated root zone and
partitions water into evaporative fluxes, groundwater recharge and fast lateral, storm flow generating
runoff. The latter of which is critical from the aspect of connectivity. In regions where Hortonian
overland flow (i.e. infiltration excess overland flow) is of minor importance, the term fast lateral flows
can represent, depending on the model and the area of application, different processes, such as
saturation overland flow, preferential flow, flow through shallow, high permeability soil layers or
combinations thereof. The interplay between water volumes that are stored and those that are released
laterally to the stream via fast, connected flow paths ("connectivity") is in most top-down models



described by functions between water stored in the unsaturated root zone ("soil moisture") and the
areal proportion of heterogeneous, local storage thresholds that are exceeded and thus "connected"
(Zhao et al., 1980). In other words, in those parts of a catchment where the storage threshold is
exceeded, no more additional water can be stored and additional water input in these parts of the
catchment will generate fast, lateral flows. This areal proportion of the catchment where thresholds are
exceed can alternatively be interpreted as runoff coefficient (e.g. Ponce and Hawkins, 1996; Perrin and
Andreassian, 2001; Fenicia et al., 2007; Bergström and Lindström, 2015). The idea goes back to the
variable contributing area concept, assuming that only partial areas of a catchment, where soils are
saturated and thus storage thresholds are exceeded, contribute to runoff (Hewlett, 1961; Dunne and
Black, 1970; Hewlett and Troendle, 1975). Although originally developed for catchments dominated by
saturation overland flow, the extension of the concept to subsurface connectivity, posing that surface
and subsurface connectivity are "two sides of the same coin" (McDonnell, 2013), proved highly valuable
for models such as Xinanjiang (Zhao et al., 1980), HBV (Bergström and Forsman, 1973; Bergström and
Lindström, 2015), SCS-CN (Ponce and Hawkins, 1996; Bartlett et al., 2016), FLEX (Fenicia et al., 2008) or
GR4J (Perrin and Andreassian et al., 2001), applied  in other regions, too.
The parameters of these storage excess distribution functions are typically calibrated. In spite of being
the core component of soil moisture routines in many top-down models, little effort was previously
invested to find ways to determine the shape of the functions describing the spatial heterogeneity of
storage thresholds and thus connectivity pattern at the catchment-scale directly from available data. An
important step towards understanding and quantifying connectivity pattern directly based on
observations was recently achieved by intensive experimental work in the Tenderfoot Creek catchments
in Montana, US. In their work Jencso et al. (2009) were able to show that connectivity of individual
hillslopes in their headwater catchments is highly related to their respective upslope accumulated areas.
Using this close relationship, Smith et al. (2013) successfully developed a simple top-down model with
very limited need for calibration, emphasizing the value of "enforcing field-based limits on model
parameters" (Smith et al., 2016). Based on hydrological landscape analysis, Savenije (2010) suggested
that as topographical features are frequently linked to distinct hydrological functional traits, they may
potentially be used to construct a conceptual catchment model based on a perceptual model of
hydrological units. Gharari et al., 2014 found that by imposing semi-quantitative relational constraints,
the FLEX-Topo model can dramatically reduce the need for calibration. The model has also shown to
hold considerable potential for spatial model transferability without the need for parameter re-
calibration (Gao et al., 2014a; H. Gao et al., 2016).



In many top-down models, such as Xinanjiang or HBV, connectivity is formulated in a general form as
$C_R$=f($S_U(t)$,$S_{uMax}$,β), where $C_R$ is the runoff coefficient, i.e. the proportion of the catchment generating
runoff, $S_U(t)$ is the catchment water content in the unsaturated root zone at any time $t$, $S_{uMax}$ is a scale
parameter representing the total storage capacity in the unsaturated root zone and β is a shape
parameter, representing the spatial distribution of heterogeneous storage capacities in the unsaturated
root zone. In a recent development, several studies suggest that $S_{uMax}$ can be robustly and directly
inferred long term water balance data, by the Mass Curve Technique (MCT), without the need for
further calibration (Gao et al., 2014; de Boer-Euser et al., 2016; Nijzink et al., 2016). This leaves shape
parameter β as only free calibration parameter for soil moisture routines of that form.
Topography is often the dominant driver of water movement caused by prevailing hydraulic gradients.
More crucially, topography usually provides an integrating indicator for hydrological behavior, since
topography is usually closely related with other landscape elements, such as soil vegetation climate and
even geology (Seibert et al., 2007; Savenije, 2010; Rempe and Dietrich, 2014; Gao et al., 2014b; Maxwell
and Condon, 2016; Gomes, 2016). The Height Above the Nearest Drainage (HAND; Rennó et al., 2008;
Nobre et al., 2011; Gharari et al., 2011), which can be computed from readily available digital elevation
models, could potentially provide first order estimates of groundwater depth , as there is some
experimental evidence that with increasing HAND, groundwater depths similarly increase (e.g. Haria and
Shand, 2004; Martin et al., 2004; Molenat et al., 2005, 2008; Shand et al., 2005; Condon and Maxwell,
2015; Maxwell and Condon, 2016). HAND can be interpreted as a proxy of the hydraulic head and is thus
potentially more hydrologically informative than the topographic elevation above sea level (Nobre et al.,
2011). Compared with the Topographic Wetness Index (TWI) in TOPMODEL, HAND is an explicit measure
of a physical feature linking terrain to water relative to the potential energy for local drainage (Nobre et
al., 2011). More interestingly, topographic structure emerges as a powerful force determining rooting
depth under a given climate or within a biome, revealed by ecological observations in global scale (Fan
et al., 2017). This leads us to think from ecological perspective to use the topographic information as an
indicator for root zone spatial distribution without calibrating the β, and coupling it with the MCT
method to estimate the $S_{uMax}$, eventually create a calibration-free runoff generation module.
In this study we are therefore going to test the hypotheses that: (1) HAND can be linked to the spatial
distribution of storage capacities (β) and therefore can be used to develop a new runoff generation
module; (2) the distribution of storage capacities determined by HAND contains different information
than the topographic wetness index; (3) the estimates of β together with water balance-based estimates



of $S_{uMax}$ allow the formulation of calibration-free parameterizations of soil moisture routines in top-down
models directly based on observations. All these hypotheses will be tested firstly in a small data-rich
experimental catchment (the Bruntland Burn catchment in Scotland), and then apply the model to a wide
range of larger catchments (MOPEX, Model Parameter Estimation Experiment).
This paper is structured as follows. In the Methods section, we describe two of our proposed modules, i.e.
HSC (HAND-based Storage Capacity curve) and HSC-MCT, and two benchmark models (HBV, TOPMODEL).
This section also includes the description of other modules (i.e. interception, evaporation and routing) in
rainfall-runoff modelling, and the methods for model evaluation, calibration and validation. The Dataset
section reviews the empirically-based knowledge of the Bruntland Burn catchment in Scotland and the
hydrometerological and topographic datasets of MOPEX catchments in the US for model comparison. The
Results section presents the model comparison results. The Discussion section interprets the relation
between rainfall-runoff processes and topography, catchment heterogeneity and simple model, and the
implications and limitations of our proposed modules. The conclusions are briefly reviewed in the
Summary and Conclusions section.

## 2   Methods

Based on our perceptual model that SEF is the dominant runoff generation mechanism in most cases, we
developed the HAND-based Storage Capacity curve (HSC) module. Subsequently, estimating the
parameter of root zone storage capacity ($S_{uMax}$) by the Mass Curve Techniques (MCT) without calibration,
the HSC-MCT was developed. In order to assess the performance of our proposed modules, two widely-
used runoff generation modules, i.e. HBV power function and TOPMODEL module, were set as
benchmarks. Other modules, i.e. interception, evaporation and routing, are kept with identical structure
and parameterization for the four rainfall-runoff models (HBV, TOPMODEL, HSC, HSC-MCT, whose names
are from their runoff generation modules), to independently diagnose the difference among runoff
generation modules (Clark et al., 2008; 2010).

### 2.1   Two benchmark modules

**HBV power function**

The HBV runoff generation module applies an empirical power function to estimate the nonlinear
relationship between the runoff coefficient and soil moisture (Bergström and Forsman, 1973; Bergström
and Lindström, 2015). The function is written as:





$$A_s = (\frac{S_u}{S_{uMax}})^\beta \qquad (1)$$

Where $A_s$ (-) represents the contributing area, which equals to the runoff coefficient of a certain rainfall
event; $S_u$ (mm) represents the averaged root zone soil moisture; $S_{uMax}$ (mm) is the averaged root zone
storage capacity of the studied catchment; $\beta$ (-) is the parameter determining the shape of the power
function. The prior range of $\beta$ can be from 0.1 to 5. The $S_u$ - $A_s$ has a linear relation while $\beta$ equals to 1. And
the shape becomes convex while the $\beta$ is less than 1, and the shape turns to concave while the $\beta$ is larger
than 1. In most situations, $S_{uMax}$ and $\beta$ are two free parameters, cannot be directly measured at the
catchment scale, and need to be calibrated based on observed rainfall-runoff data.
**TOPMODEL module**
The TOPMODEL assumes topographic information captures the runoff generation heterogeneity at
catchment scale, and the TWI is used as an index to identify rainfall-runoff similarity (Beven and Kirkby,
1979; Sivapalan et al., 1997). Areas with similar TWI values are regarded as possessing equal runoff
generation potential. More specifically, the areas with larger TWI values tend to be saturated first and
contribute to SEF; but the areas with lower TWI values need more water to reach saturation and generate
runoff. The equations are written as follow:
$$D_i = \overline{D} + S_{uMax}(\overline{I_{TW}} - I_{TW_i}) \qquad (2)$$

$$\overline{D} = S_{uMax} - S_u \qquad (3)$$

$$A_s = \sum A_{s\_i}; \quad \text{while} \ D_i < 0 \qquad (4)$$

Where $D_i$ (mm) is the local storage deficit below saturation at specific location ($i$); $\overline{D}$ (mm) is the averaged
water deficit of the entire catchment (Equation 2), which equals to ($S_{uMax}$ - $S_u$), as shown in Equation 3. $I_{TWi}$
is the local $I_{TW}$ value. $\overline{I_{TW}}$ is the averaged TWI of the entire catchment. Equation 2 means in a certain soil
moisture deficit condition for the entire catchment ($\overline{D}$), the soil moisture deficit of a specific location ($D_i$),
is determined by the catchment topography ($I_{TW}$ and $I_{TWi}$), and the root zone storage capacity ($S_{uMax}$).
Therefore, the areas with $D_i$ less than zero are the saturated areas ($A_{s\_i}$), equal to the contributing areas.
The integration of the $A_{s\_i}$ areas ($A_s$), as presented in Equation 4, is the runoff contributing area, which
equals to the runoff coefficient of that rainfall event.



Besides continuous rainfall-runoff calculation, Equations 2-4 also allow us to obtain the contributing area
($A_s$) from the estimated relative soil moisture ($S_u/S_{uMax}$), and then map it back to the original TWI map,
which makes it possible to test the simulated contributing area by field measurement. It is worth
mentioning that the TOPMODEL in this study is a simplified version, and not identical to the original one,
which combines the saturated and unsaturated soil components.

## 2.2 HSC module

Hydrological models are human constructs that simplify the larger reality of hydrological processes
(Savenije, 2009), and assumptions are inevitable (Neuweiler and Helmig, 2017). In this study, we assume
1) SEF is the dominant runoff generation mechanism, while SOF and SSF cannot be distinguished; 2) the
local root zone storage capacity has a positive and linear relationship with HAND, from which we can
derive the spatial distribution of the root zone storage capacity; 3) HAND contours are parallel to each
other in runoff generation. We believe that rainfall firstly feeds local soil moisture deficit, and no runoff
can be generated before areas being subsequently saturated and water moving downslope. And after
being saturated and connected with the channel network, either the cascade or parallel model structure
does not impact on runoff generation. So this parallel model structure not only simplifies our simulation,
it is also very likely closer to reality (Savenije, 2010).
Figure 1 shows the perceptual HSC module, in which we simplified the complicated 3-D topography of a
real catchment into a 2-D simplified hillslope. And then derive the distribution of root zone storage
capacity, based on topographic analysis and the second assumption as mentioned in the preceding
paragraph. Figure 2 shows the approach to derive the $S_u$-$A_s$ relation, which are detailed as follows.
I.   **Generate HAND map.** The HAND map of study catchment can be generated from Digital Elevation
Model (DEM) (Gharari et al., 2011). The stream initiation threshold area is a crucial parameter,
determining the perennial river channel network (Montgomery and Dietrich, 1989; Hooshyar et
al., 2016), and significantly impacting the HAND values.
II.  **Generate normalized HAND distribution curve.** Firstly, sort the HAND values of grid cells in
ascending order. Secondly, the sorted HAND values were evenly divided into $n$ bands (e.g. 20
bands in this study), to make sure each HAND band has similar area. The averaged HAND value of
each band is regarded as the HAND value of that band. Thirdly, normalize the HAND bands, and
then plot the normalized HAND distribution curve (Figure 1b).
III. **Distribute $S_{uMax}$ to each HAND band ($S_{uMax\_i}$).** As assumed, the normalized storage capacity of each
HAND band ($S_{uMax\_i}$) increases with HAND value (Figure 1c). Based on this assumption, the




unsaturated root zone storage capacity ($S_{uMax}$) can be distributed to each HAND band as $S_{uMax\_i}$

(Figure 2a). It is worth noting that $S_{uMax}$ needs to be calibrated in the HSC module, but free of

calibration in the HSC-MCT module.

IV.    **Derive the $S_u$ - $A_s$ curve.** With the number of $s$ saturated HAND bands (Figure 2a-c), the soil

moisture ($S_u$) can be obtained by Equation 5; and saturated area proportion ($A_s$) can be obtained

by Equation 6.

$$S_{u} = \frac{1}{n}\left[\sum_{i=1}^{s} S_{uMax\_i} + S_{uMax\_s}(n - s)\right] \tag{5}$$

$$A_{s} = \frac{s}{n} \tag{6}$$

Where $S_{uMax\_s}$ is the maximum $S_{uMax\_i}$ of all the saturated HAND bands. Subsequently, the $A_s$ - $S_u$

curve can be derived, and shown in Figure 2d.

The SEF mechanism assumes that only the saturated areas generate runoff, therefore the proportion of
saturation area is equal to the runoff coefficient of that rainfall-runoff event. Based on the $S_u$-$A_s$ curve in
Figure 2d, generated runoff can be calculated from root zone moisture ($S_u$). The HSC module also allows
us to map out the fluctuation of saturated areas by the simulated catchment average soil moisture. For
each time step, the module can generate the simulated root zone moisture for the entire basin ($S_u$). Based
on the $S_u$-$A_s$ relationship (Figure 2d), we can map $S_u$ back to the saturated area proportion ($A_s$) and then
visualize it in the original HAND map. Based on this conceptual model, we developed the computer
program and created a procedural module. The technical roadmap can be found in Figure 3.
## 2.3  HSC-MCT module
The $S_{uMax}$ is an essential parameter in various hydrological models (e.g. HBV, Xinanjiang, GR4J), which
determines the long-term partitioning of rainfall into infiltration and runoff. Gao et al., 2014a found that
$S_{uMax}$ represents the adaption of ecosystems to local climate. Ecosystems may design their $S_{uMax}$ based on
the precipitation pattern and its water demand. The storage is neither too small to avoid mortality in dry
seasons, nor too large to consume excessive energy and nutrients. Based on this assumption, we can
estimate the $S_{uMax}$ without calibration, by the MCT method, from climatological and vegetation
information. More specifically, the average annual plant water demand in the dry season ($S_R$) is
determined by the water balance and the vegetation phenology, i.e. precipitation, runoff and seasonal
NDVI. Subsequently, based on the annual $S_R$, the Gumbel distribution (Gumbel, 1935), frequently used for
estimating hydrological extremes, was used to standardize the frequency of drought occurrence. $S_{R20y}$, i.e.
the root zone storage capacity required to overcome a drought once in 20 years, is used as the proxy for





$S_{uMax}$ due to the assumption of a "cost" minimization strategy of plants as we mentioned above (Milly,
1994), and the fact that $S_{R20y}$ has the best fit with $S_{uMax}$ (Gao et al., 2014a).
Eventually, with the MCT approach to estimate $S_{uMax}$ and the HSC curve to represent the root zone storage
capacity spatial distribution, the HSC-MCT runoff generation module is created, without free parameters.
It is worth noting that both the HSC-MCT and HSC modules are based on the HAND derived $S_u$-$A_s$ relation,
and their distinction lays in the methods to obtain $S_{uMax}$. So far, the HBV power function module has 2 free
parameters ($S_{uMax}$, β). While the TOPMODEL and the HSC both have one free parameter ($S_{uMax}$). Ultimately
the HSC-MCT has no free parameter.

## 2.4    Interception, evaporation and routing modules

Except for the runoff generation module in the root zone reservoir ($S_{UR}$), we need to consider other
processes, including interception ($S_{IR}$) before the $S_{UR}$ module, evaporation from the $S_{UR}$ and the response
routine ($S_{FR}$ and $S_{SR}$) after runoff generation from $S_{UR}$ (Figure 4). Precipitation is firstly intercepted by
vegetation canopies. In this study, the interception was estimated by a threshold parameter ($S_{iMax}$), set to
2 mm (Gao et al., 2014a), below which all precipitation will be intercepted and evaporated (Equation 9)
(de Groen and Savenije, 2006). For the $S_{UR}$ reservoir, we can either use the HBV beta-function (Equation
12), the runoff generation module of TOPMODEL (Equation 2-4) or the HSC module (Section 2.3) to
partition precipitation into generated runoff ($R_u$) and infiltration. The actual evaporation ($E_a$) from the soil
equals to the potential evaporation ($E_p$), if $S_u/S_{uMax}$ is above a threshold ($C_e$), where $S_u$ is the soil moisture
and $S_{uMax}$ is the catchment averaged storage capacity. And $E_a$ linearly reduces with $S_u/S_{uMax}$, while $S_u/S_{uMax}$
is below $C_e$ (Equation 13). The $E_p$ can be calculated by the Hargreaves equation (Hargreaves and Samani,
1985), with maximum and minimum daily temperature as input. The generated runoff ($R_u$) is further split
into two fluxes, including the flux to the fast response reservoir ($R_f$) and the flux to the slow response
reservoir ($R_s$), by a splitter ($D$) (Equation 14, 15). The delayed time from rainfall peak to the flood peak is
estimated by a convolution delay function, with a delay time of $T_{lagF}$. Subsequently, the fluxes into two
different response reservoirs ($S_{FR}$ and $S_{SR}$) were released by two linear equations between discharge and
storage (Equation 19, 21), representing the fast response flow and the slow response flow mainly from
groundwater reservoir. The two discharges ($Q_f$ and $Q_s$) generated the simulated streamflow ($Q_m$). The
model parameters are shown in Table 1, while the equations are given in Table 2. More detailed
description of the model structure can be referred to Gao et al., 2014b and 2016. It is worth underlining
that the only difference among the benchmark HBV type, TOPMODEL type, the HSC and the HSC-MCT



models is their runoff generation modules. Eventually, there are 7 free parameters in HBV model, 6 in
TOPMODEL and HSC model, and 5 in the HSC-MCT model.

## 2.5    Model evaluation, calibration, validation and models comparison

Two objective functions were used to evaluate model performance, since multi-objective evaluation is a
more robust approach to quantifying model performance with different criteria than a single one. The
Kling-Gupta efficiency (Gupta *et al.*, 2009) ($I_{KGE}$) was used as the criteria to evaluate model performance
and as an objective function for calibration. The equation is written as:
$$I_{KGE} = 1 - \sqrt{(r-1)^2 + (\alpha-1)^2 + (\varepsilon-1)^2}$$
(7)

Where $r$ is the linear correlation coefficient between simulation and observation; $\alpha$ ($\alpha = \sigma_m / \sigma_o$) is a
measure of relative variability in the simulated and observed values, where $\sigma_m$ is the standard deviation
of simulated streamflow, and $\sigma_o$ is the standard deviation of observed streamflow; $\varepsilon$ is the ratio between
the average value of simulated and observed data. And the $I_{KGL}$ ($I_{KGE}$ of the logarithmic flows) (Fenicia et
al., 2007; Gao et al., 2014b) is used to evaluate the model performance on baseflow simulation. Since the
response module, determined the baseflow simulation and $I_{KGL}$, is kept the same for all four models, thus
only the $I_{KGE}$ results are presented in the results.
A multi-objective parameter optimization algorithm (MOSCEM-UA) (Vrugt et al., 2003) was applied for
the calibration. The parameter sets on the Pareto-frontier of the multi-objective optimization were
assumed to be the behavioral parameter sets and can equally represent model performance. The
averaged hydrograph obtained by all the behavioral parameter sets were regarded as the simulated result
of that catchment for further studies. The number of complexes in MOSCEM-UA were set as the number
of parameters (7 for HBV, 6 for TOPMODEL and the HSC model, and 5 for HSC-MCT model), and the
number of initial samples was set to 210 and a total number of 50000 model iterations for all the
catchment runs. For each catchment, the first half period of data was used for calibration, and the other
half was used to do validation.
In module comparison, we defined three categories: if the difference of $I_{KGE}$ of model A and model B in
validation is less than 0.1, model A and B are regarded as "equally well". If the $I_{KGE}$ of model A is larger
than model B in validation by 0.1 or more, model A is regarded as outperforming model B. If the $I_{KGE}$ of
model A is less than model B in validation by -0.1 or less, model B is regarded to outperform model A.



## 3  Dataset

### 3.1  The Bruntland Burn catchment

The 3.2 km² Bruntland Burn catchment (Figure 5), located in north-eastern Scotland, was used as a
benchmark study to test the models performance based on a rich data base of hydrological measurements.
The Bruntland Burn is typical many upland catchments in North West Europe (e.g. Birkel et al., 2010),
namely a combination of steep and rolling hillslopes and over-widened valley bottoms due to the glacial
legacy of this region. The valley bottom areas are covered by deep (in parts > 30m) glacial drift deposits
(e.g. till) containing a large amount of stored water superimposed on a relatively impermeable granitic
solid geology (Soulsby et al., 2016). Peat soils developed (> 1m deep) in these valley bottom areas, which
remain saturated throughout most of the year with a dominant near-surface runoff generation
mechanism delivering runoff quickly via micro-topographical flow pathways connected to the stream
network (Soulsby et al., 2015). Brown rankers, peaty rankers and peat soils are responsible for a flashy
hydrological regime driven by saturation excess overland flow, while humus iron podzols on the hillslopes
do not favor near-surface saturation but rather facilitate groundwater recharge through vertical water
movement (Tetzlaff et al., 2014). Land-use is dominated by heather moorland, with smaller areas of rough
grazing and forestry on the lower hillslopes. Its annual precipitation is 1059 mm, with the summer months
(May-August) generally being the driest (Ali et al., 2013). Snow makes up less than 10% of annual
precipitation and melts rapidly below 500m. The evapotranspiration is around 400 mm per year and
annual discharge around 659 mm.
The LiDAR-derived DEM map with 2m resolution shows elevation ranging from 250m to 539m (Figure 5).
There are 7 saturation area maps (Figure 6) (May 2, July 2, August 4, September 3, October 1, November
26, in 2008, and January 21, in 2009), measured directly by field mapping (Ali et al., 2013), and a global
positioning system (GPS) was used to delineate the boundary of saturation areas. These saturation area
maps revealed a dynamic behavior of expanding and contracting areas connected to the stream network
that were used as a benchmark test for the HSC module.

### 3.2  MOPEX dataset

The MOPEX dataset was collected for a hydrological model parameter estimation experiment (Duan et al.,
2006; Schaake et al., 2006), containing 438 catchments in the CONUS (Contiguous United States). The
dataset contains the daily precipitation, daily maximum and minimum air temperature, and daily
streamflow. The longest time series range from 1948 to 2003. 323 catchments were used in this study,





with areas between 67 and 10,329 km$^2$, and excluding the catchments with data records <30 years,
impacted by snowmelt or with extreme arid climate (aridity index $E_p/P > 2$). The daily streamflow was used
to calibrate the free parameters, and validate the models. The Digital Elevation Model (DEM) of the
CONUS in 90m resolution was download from the Earth Explorer of United States Geological Survey (USGS,
http://earthexplorer.usgs.gov/).

## 361   4   Results of the Bruntland Burn

### 362   4.1   Topography analysis

The TWI map of the BB (Figure 5) was generated from its DEM. Overall, the TWI map, ranging from -0.4
to 23.4, mainly differentiates the valley bottom areas with the highest TWI values from the steeper slopes.
This is probably caused by the fine resolution of the DEM map in 2 m, since previous research found the
sensitivity of TWI to DEM resolution (Sørensen and Seibert, 2007). From the TWI map, the frequency
distribution function and the accumulative frequency distribution function can be derived (Figure 7), with
one unit of TWI as interval.
The generated HAND map, derived also from the DEM, is shown in Figure 5, with HAND values ranging
from 0m to 234m. Since HAND is sensitive to the definition of the perennial channel, which is highly
impacted by the stream initiation threshold area (Montgomery and Dietrich, 1989; Gharari et al., 2011;
Hooshyar et al., 2016). The start area was chosen as 40ha to maintain a close correspondence with
observed stream network. Based on the HAND map, we can derive the $S_u$-$A_s$ curve (Figure 7) by analyzing
the HAND map with the method in Section 2.3.

### 375   4.2   Model performance

The observed and simulated hydrographs of three models (HBV, TOPMODEL, and HSC) in 2008 are shown
in Figure 8. We found all the three models can perform well to reproduce the observed hydrograph. The
$I_{KGE}$ of the three models are all around 0.66, which is largely in line with other studies from the BB (Birkel
et al, 2010; 2014). Since the measured rainfall-runoff time series only last from 2008 to 2014, which is too
short to estimate the $S_{R20y}$ (proxy for $S_{uMax}$) by MCT approach (which needs long-term hydro-
meteorological observation data,) the HSC-MCT model was not applied to the catchment.
The normalized relative soil moisture of the three model simulations are presented in Figure 8. Their
temporal fluctuation patterns are comparable. Nevertheless, the simulated soil moisture by TOPMODEL
has a larger variation, compared with HBV and HSC (Figure 8).



Figure 7 shows the calibrated power curve from HBV (beta=0.98) with the $S_u$-$A_s$ curve obtained from the
HSC module. We found the two curves are largely comparable, especially while the relative soil moisture
is low. This result demonstrates that for the BB with glacial drift deposits and combined terrain of steep
and rolling hillslopes and over-widened valley bottoms, the HBV power curve can essentially be derived
from the $S_u$-$A_s$ curve of HSC module merely by topographic information without calibration.

## 4.3  Contributing area simulation

The observed saturation area and the simulated contributing area from both TOPMODEL and the HSC are
shown in Figure 6, 8, 9. We found although both modules overestimated the contributing areas, they can
capture the temporal variation. For example, the smallest saturated area both observed and simulated
occurred on July-02-2008, and the largest saturated area both occurred on January-21-2009. Comparing
the estimated contributing area of TOPMODEL with the HSC module, we found the results of the HSC
correlates better ($R^2$=0.60) with the observed saturated areas than TOPMODEL ($R^2$=0.50) (Figure 9). For
spatial patterns, the results of the HSC module are also more closely comparable with the observed
saturated areas than TOPMODEL (Figure 6). Based on these results benchmarking the HSC module with
observed saturated area maps, we proceeded to test HSC for a wide range of climatically and
geomorphologically different catchments across the US.

# 5  Results from the MOPEX catchments

## 5.1  Topography analysis of the Contiguous US and 323 MOPEX catchments

To delineate the TWI map for the CONUS, the depressions of the DEM were firstly filled with a threshold
height of 100m (recommended by Esri). The slope map, i.e. the steepest local slope, was generated by the
filled DEM, and the flow direction map was also derived from the filled DEM using the D8 algorithm
(O'Callaghan and Mark, 1984; Jenson and Domingue, 1988). Subsequently, the flow accumulation map
could be generated from the flow direction map. The accumulated upslope area (A), obtained from the
flow accumulation map, was then divided by an estimate of the contour length (L), which is related to the
flow direction map, to provide the local upslope area draining through a certain point per unit contour
length (a = A/L). With the definition of TWI as ln(a/tan β), the TWI map of the CONUS is produced (Figure
S1). From the TWI map of the CONUS, we clipped the TWI maps for the 323 MOPEX catchments with their
catchment boundaries. And then the TWI frequency distribution and the accumulated frequency
distribution of the 323 MOPEX catchments (Figure S2), with one unit of TWI as interval, were derived
based on the 323 TWI maps.





The HAND map (Figure 10) was also generated from the filled DEM, with the input of the flow direction
and flow accumulation maps. Specifically, the perennial river network was obtained, based on the flow
accumulation map, by setting the stream initiation area threshold of 500 grid cells (4.05 km$^2$), which fills
in the range of stream initiation thresholds reported by others (e.g. Colombo et al., 2007; Moussa, 2008,
2009). In the end, HAND was then calculated from the elevation of each raster cell above nearest grid cell
flagged as stream cell following the flow direction (Gharari et al., 2011).
In Figure 10, it is shown that the regions with large HAND values are located in Rocky Mountains and
Appalachian Mountains, while the Great Plains has smaller HAND values. Interestingly, the Great Basin,
especially in the Salt Lake Desert, has small HAND values, illustrating its low elevation above the nearest
drainage, although their elevations above seas level are high. From the CONUS HAND map, we clipped the
HAND maps for the 323 MOPEX catchments with their catchment boundaries. We then plot their HAND-
area curves, following the procedures of II-IV in Section 2.2. Figure 11a shows the normalized HAND
profiles of the 323 catchments.
Based on the HAND profiles and the Step V in Section 2.2, we derived the normalized storage capacity
distribution for all catchments (Figure 11b). Subsequently, the root zone moisture and saturated area
relationship ($A_s$-$S_u$) can be plotted by the method in Step VI of Section 2.2. Lastly, reversing the curve of
$A_s$-$S_u$ to $S_u$-$A_s$ relation (Figure 11c), the latter one can be implemented to simulate runoff generation by
soil moisture. Figure 11c interestingly shows that in some catchments, there is almost no threshold
behavior between rainfall and runoff generation, where the catchments are covered by large areas with
low HAND values and limited storage capacity. Therefore, when rainfall occurs, wetlands response quickly
and generate runoff without a precipitation–discharge threshold relationship characteristic of areas with
higher moisture deficits. This is similar to the idea of FLEX-Topo where the storage capacity is distinguished
between wetlands and hillslopes, and on wetlands, with low storage capacity, where runoff response to
rainfall is almost instantaneous.
## 5.2    Model performance
Overall, the performance of the two benchmark models, i.e. HBV and TOPMODEL, for the MOPEX data
(Figure 12) is comparable with the previous model comparison experiments, conducted with four rainfall-
runoff models and four land surface parameterization schemes (Duan et al., 2006; Ye et al., 2014). The
median value of $I_{KGE}$ of the HBV type model is 0.61 for calibration in the 323 catchments (Figure 12), and
averaged $I_{KGE}$ in calibration is 0.62. In validation, the median and averaged values of $I_{KGE}$ are kept the same
as calibration. The comparable performance of models in calibration and validation demonstrates the



robustness of benchmark models and the parameter optimization algorithm (i.e. MOSCEM-UA). The
TOPMODEL improves the median value of $I_{KGE}$ from 0.61 (HBV) to 0.67 in calibration, and from 0.61 (HBV)
to 0.67 in validation. But the averaged values of $I_{KGE}$ for TOPMODEL are slightly decreased from 0.62 (HBV)
to 0.61 in both calibration and validation. The HSC module, by involving the HAND topographic
information without calibrating the β parameter, improves the median value of $I_{KGE}$ to 0.68 for calibration
and 0.67 for validation. The averaged values of $I_{KGE}$ in both calibration and validation are also increased to
0.65, comparing with HBV (0.62) and TOPMODEL (0.61). Furthermore, Figure 12 demonstrates that,
comparing with the benchmark HBV and TOPMODEL, not only the median and averaged values were
improved by the HSC module, but also the 25[th] and 75[th] percentiles and the lower whisker end, all have
been dramatically improved. These results indicate the HSC module improved model performance to
reproduce hydrograph, and simultaneously maintaining model robustness and consistency.
Additionally, for the median $I_{KGE}$ value, the HSC-MCT leads to an improvement from 0.61 (HBV) to 0.65 in
calibration, and from 0.61 (HBV) to 0.64 in validation, but not as well performed as TOPMODEL (0.67 for
calibration and validation). For the averaged $I_{KGE}$ values, they were slightly reduced from 0.62 (HBV) and
0.61 (TOPMODEL) to 0.59 for calibration and validation. Although the HSC-MCT did not perform as well
as the HSC module, considering there is no free parameters to calibrate, the median $I_{KGE}$ value of 0.64
(HBV is 0.61) and averaged $I_{KGE}$ of 0.59 (TOPMODEL is 0.61) are quite acceptable. In addition, the 25[th] and
75[th] percentiles and the lower whisker end of the HSC-MCT model are all improved compared to the HBV
model. Moreover, the largely comparable results between the HSC and the HSC-MCT modules
demonstrate the feasibility of the MCT method to obtain the $S_{uMax}$ parameter and the potential for HSC-
MCT to be implemented in prediction of ungauged basins (PUB, cf. Sivapalan et al., 2003; Blöschl et al.,
2013; Hrachowitz et al., 2013). Since the response routines determining the baseflow simulation of the
four models are exactly the same, the results of $I_{KGL}$ as an indicator to evaluate baseflow are not presented.
Figure 13 shows the spatial comparisons of the HSC and HSC-MCT models with the two benchmark models.
We found that the HSC performs "equally well" as HBV (the difference of $I_{KGE}$ in validation ranges -0.1 ~
0.1) in 88% catchments, and in the remaining 12% of the catchments the HSC outperforms HBV (the
improvement of $I_{KGE}$ in validation is larger than 0.1). In not a single catchment did the calibrated HBV
outperform the HSC. From the spatial comparison, we found that the catchments, where the HSC model
performed better are mostly located in the Great Plains, with modest sloping (4.0 degree), while the other
catchments have average slope of 8.1 degree. Comparing the HSC model with TOPMODEL, we found in
91% of the catchments that the two models have approximately equal performance. In 8% of the



catchments, the HSC model outperformed TOPMODEL. Only in 1% of the catchments (two in Appalachian Mountain and one in the Rocky Mountain in California), TOPMODEL performed better. From spatial analysis, we found the HSC outperformed catchments have flat terrain (2.3 degree) with moderate averaged HAND value (26m), while the TOPMODEL outperformed catchments have steep hillslope (19 degree) with large averaged HAND value (154m).

Without calibration of $S_{uMax}$, as expected, the performance of HSC-MCT module slightly deteriorates (Figure 12). In comparison with HBV, the outperformance reduced from 12% (HSC) to 4% (HSC-MCT), the approximately equal-well simulated catchments dropped from 88% to 79%, and the inferior performance increased from 0% to 17%. Also, in comparison with TOPMODEL, the better performance dropped from 8% (HSC model) to 7% (HSC-MCT model), the approximately equal catchments reduced from 91% to 72%, and the inferior performance increased from 1% to 21%. The inferiority of the HSC-MCT model is probably caused by the uncertainty of the MCT method for different ecosystems which have different survival strategies and use different return periods to bridge critical drought periods. By using ecosystem dependent return periods, this problem could be reduced (Wang-Erlandsson et al., 2016).

To further explore the reason for the better performance of the HSC approach, we selected the 08171000 catchment in Texas (Figure 13), in which both the HSC module and the HSC-MCT module outperformed the two benchmark modules to reproduce the observed hydrograph (Figure S3). The HBV model dramatically underestimated the peak flows, with $I_{KGE}$ as 0.54, while TOPMODEL significantly overestimated the peak flows, with $I_{KGE}$ as 0.30. The HSC-MCT model improved the $I_{KGE}$ to 0.71, and the HSC model further enhanced $I_{KGE}$ to 0.74.

Since the modules of interception, evaporation and routing are identical for the four models, the runoff generation modules are the key to understand the difference in model performance. Figure S4 shows the HBV β curve and the $S_u$-$A_s$ curve of the HSC model, as well the TWI frequency distribution. We found that with a given $S_u/S_{uMax}$, the HBV β function generates less contributing area than the HSC model, which explains the underestimation of the HBV model. In contrast, TOPMODEL has a sharp and steep accumulated TWI frequency curve. In particular, the region with TWI=8 accounts for 40% of the catchment area, and over 95% of the catchment areas are within the TWI ranging from 6 to 12. This indicates that even with low soil moisture content ($S_u/S_{uMax}$), the contributing area by TOPMODEL is relatively large, leading to the sharply increased peak flows for all rainfall events.



## 6   Discussion

### 6.1   Rainfall-runoff processes and topography

In hillslope and catchment hydrology, the partitioning of precipitation into runoff and evaporation is a fundamental function in virtually all hydrological models. Bucket-type models (e.g. HBV and Xinanjiang), as one of the most widely used group of conceptual models, typically adopt two parameters to determine runoff generation. One is the root zone storage capacity ($S_{uMax}$) and the other is the shape parameter (i.e. $\beta$) determining the relation between root zone moisture and runoff generation. In long-term water balance studies, climate plays a key role in determining the storage capacity and the partition between evaporation and runoff (Budyko, 1971; Wang and Tang, 2014; Gao et al., 2014a). But for specific events, the key question is how antecedent soil moisture impacts runoff generation. Field studies support that in many mildly sloping catchments, SEF is the dominant runoff generation mechanism (Sklash and Farvolden, 1979; Burt and McDonnell, 2015). Therefore, it is essential to determine the temporal variability of the saturated area (equal to the contributing area and the runoff coefficient of a specific rainfall-runoff event), to calculate runoff generation. Linking the runoff contributing area to topography is not a new insight in rainfall-runoff modelling. TOPMODEL is an elegant pioneering model allowing us to understand the interaction between topography and connectivity. In this study, the HSC module uses a relatively new topographic index (i.e. HAND) to identify hydrological similarity.

We applied a novel approach to derive the relationship between soil moisture storage and the saturated area from HAND. The areas with relatively low HAND values are saturated earlier than areas with higher HAND values, due to the larger storage capacity in high HAND locations. The outperformance of the HSC model over the benchmark HBV and TOPMODEL in modestly sloping catchments indicates that the HSC module has a higher realism than the calibrated beta-function of the HBV model and the TWI of TOPMODEL in these regions. Very interestingly, Fan et al., (2017) presented a global synthesis of 2,200 root observations of >1000 species, and revealed the systematic variation of rooting depth along HAND (Fig.1, in Fan et al., 2017). Since rooting depth can be translated to root zone storage capacity through combination with soil plant-available water (Wang-Erlandsson et al., 2016). This large sample dataset, from ecological perspective, provides a strong support for the assumption of the HSC model on modest slopes, i.e. the increase of root zone storage capacity with HAND. More interestingly, on excessively drained uplands, rooting depth does not follow the same role, with shallow depth and limited to rain infiltration (Fig.1, in Fan et al., 2017). This could explain the inferior performance of HSC model to TOPMODEL in three MOPEX catchments (averaged HAND is 154 m) with excessively drained uplands,



where Hortonian overland flow is likely the dominant mechanism, and the HSC assumption likely does not
work well.
The FLEX-Topo model (Savenije, 2010) also uses HAND information as a topographic index to distinguish
between landscape-related runoff processes, and has both similarity and differences with the HSC model.
The results of the HSC model illustrate that the riparian areas are more prone to be saturated, which is
consistent with the concept of the FLEX-Topo model. Another important similarity of the two models is
their parallel model structure. From our perspective, the parallel model structure is closer to reality. Since
before saturation, rainfall is firstly infiltrated into local storage, and water moves vertically; only after a
certain level of saturation, water starts to move laterally. But in both models it is assumed that the upslope
area has larger storage capacity, therefore the upper land generates runoff later than the lower land. In
other words, in most cases, the local storage is saturated due to the local rainfall, instead of flow from
upslope. Therefore, the local storage is an essential feature, to estimate the saturated area and runoff
generation, rather than the water coming from uphill.
The most obvious difference between the HSC and the FLEX-Topo models is the approach towards
discretization of a catchment. The FLEX-Topo model classifies a catchment into various landscapes, e.g.
wetlands, hillslopes and plateau. This discretization method requires threshold values to classify
landscapes, i.e. threshold values of HAND and slope, which leads to fixed and time-independent
proportions of landscapes. The HSC model does not require landscape classification, which reduced the
subjectivity in discretization and restricted the model complexity, as well as simultaneously allowing the
fluctuation of saturated areas (termed as wetlands in FLEX-Topo).

## 6.2   Catchment heterogeneity and simple models

Catchments exhibit a wide array of heterogeneity and complexity with spatial and temporal variations of
landscape characteristics and climate inputs. For example, the Darcy-Richards equation approach is often
consistent with point-scale measurements of matrix flow, but not for preferential flow caused by roots,
soil fauna and even cracks and fissures (Beven and Germann, 1982; Zehe and Fluehler, 2001; Weiler and
McDonnell, 2007). As a result, field experimentalists continue to characterize and catalogue a variety of
runoff processes, and hydrological and land surface modelers are developing more and more complicated
models to involve the increasingly detailed processes (McDonnell et al., 2007). However, there is still no
compelling evidence to support the outperformance of sophisticated "physically-based" models in terms
of higher equifinality and uncertainty than the simple lumped or semi-distributed conceptual models in
rainfall-runoff simulation (Beven, 1989; Orth et al., 2015).



But evidence is mounting that a catchment is not a random assemblage of different heterogeneous parts
(Sivapalan, 2009; Troch et al., 2013; Zehe et al., 2013), and conceptualising heterogeneities does not
require complex laws (Chase, 1992; Passalacqua et al., 2015). Asking questions of "why" rather than "what"
likely leads to more useful insights and a new way forward (McDonnell et al., 2007). Catchment is a
geomorphological and even an ecological system whose parts are related to each other probably due to
catchment self-organization and evolution (Sivapalan and Blöschl, 2015; Savenije and Hrachowitz, 2017).
This encourages the hope that simplified concepts may be found adequate to describe and model the
operation of the basin runoff generation process. It is clear that topography, with fractal characteristic
(Rodriguez-Iturbe and Rinaldo, 1997), is often the dominant driver of runoff, as well as being a good
integrated indicator for vegetation cover (Gao et al., 2014b), rooting depth (Fan et al., 2017), root zone
evaporation and transpiration deficits (Maxwell and Condon, 2016), soil properties (Seibert et al., 2007),
and even geology (Rempe and Dietrich, 2014; Gomes, 2016). Therefore, we argue that increasingly
detailed topographic information is an excellent integrated indicator allowing modelers to continue
systematically represent heterogeneities and simultaneously reduce model complexity. The model
structure and parametrization of both HSC and TOPMODEL are simple, but not over simplified, as they
capture probably the most dominant factor controlling runoff generation, i.e. the spatial heterogeneity of
storage capacity. Hence, this study also sheds light on the possibility of moving beyond heterogeneity and
process complexity (McDonnell et al., 2007), to simplify them into a succinct and *a priori* curve by taking
advantage of catchment self-organization probably caused by co-evolution or the principle of maximum
entropy production (Kleidon and Lorenz, 2004).
## 6.3    Implications and limitation
The calibration-free HSC-MCT runoff generation model may enhance our ability to predict runoff in
ungauged basins (Sivapalan et al., 2003; Blöschl et al., 2013; Hrachowitz et al., 2013). Hydrological models
still depend largely on observational data to feed statistical analysis and calibrate the free parameters.
This is probably not a major issue in the developed world, with abundant of comprehensive
measurements in many places, but for the developing world it requires prediction with sparse data and
fragmentary knowledge. Topographic information with high spatial resolution is freely available globally,
allowing us to implement the HSC model in global scale studies. In addition, thanks to the recent
development, testing, and validation of remote sensing evaporation products in large spatial scale (e.g.
Anderson et al., 2011; Hu and Jia, 2015), the $S_{uMax}$ estimation has become possible without in situ hydro-



meteorological measurements (Wang-Erlandsson et al., 2016). These widely-accessible datasets make the
global-scale implementation of HSC-MCT module promising.
Finally, we should not ignore the limitations of this model, although it has better performance and
modelling consistency. Firstly, the threshold area for the initiating a stream was set as a constant value
for the entire CONUS, but the variation of this value in different climate, geology and landscape classes
(Montgomery and Dietrich, 1989; Helmlinger et al., 1993; Colombo et al., 2007; Moussa, 2008) needs to
be future investigated. Secondly, the discrepancy between observed and simulated saturation area needs
to be further investigated, by utilizing more advanced field measurement and simultaneously refining the
model assumption. The overestimation of the HSC model is possibly because two runoff generation
mechanisms – SOF and the SSF occur at the same time. However, the saturated area observed by the
"squishy boot" method (Ali et al., 2013), probably only distinguished the areas where SOF occurred.
Subsurface stormflow, also contributes to runoff but without surface runoff, cannot be observed by the
"squishy boot" method. Thus, this mismatch between simulation and observation probably leads to this
saturated area overestimation. Another interpretation might be the different definition of "saturation".
The observed saturated areas are places where 100% of soil pore volume is filled by water. But the
modelled saturation areas are located where soil moisture is above field capacity, and not necessarily 100%
filled with water, which probably also results in overestimation of saturated areas.

## 7   Summary and conclusions

In this study, we developed a simple, calibration-free hydrological module based on HAND information,
which is an excellent indictor of hydrologic similarity and a physically-based index linking terrain with
hydraulic gradient at the hillslope and catchment scales. We assumed that the local storage capacity is
closely linked to HAND. Based on this assumption and the HAND spatial distribution pattern, the soil
moisture ($S_u$) - saturated area ($A_s$) relation for each catchment was derived, which was used to estimate
the $A_s$ of specific rainfall event based on continuous calculation of $S_u$. Subsequently, based on the $S_u$-$A_s$
relation, the HAND-based Storage Capacity curve (HSC) module was developed. Then, applying the mass
curve technique (MCT) approach (Gao et al., 2014a), we estimated the root zone storage capacity ($S_{uMax}$)
from observed hydro-climatological and vegetation data, and coupled it with HSC to create the
calibration-free HSC-MCT module, in which the $S_{uMax}$ was obtained by MCT, and the $S_u$-$A_s$ relation was
obtained by HSC. The HBV beta-function and TWI-based TOPMODEL were used as two benchmark
modules to test the performance of HSC and HSC-MCT on both hydrograph simulation and ability to
reproduce the contributing area, which was measured for different hydrometeorological conditions in the



Bruntland Burn catchment in Scotland. Subsequently, 323 MOPEX catchments in the US were used as a
large sample hydrological study to further validate the effectiveness of our proposed runoff generation
modules.
In the BB exploratory study, we found that the HSC, HBV and TOPMODEL performed comparably well to
reproduce the observed hydrograph. Interestingly, the $S_u$-$A_s$ curves of HSC and HBV are largely comparable,
which illustrates the HSC curve can be used as a proxy for the HBV beta-function. Comparing the estimated
contributing area of TOPMODEL with the HSC module, we found that the results of the HSC module
correlate better ($R^2$=0.60) with the observed saturated areas compared to TOPMODEL ($R^2$=0.50). This
likely indicates that HAND maybe a better indicator to distinguish hydrological similarity than TWI.
For the 323 MOPEX catchments, HSC improved the averaged validation value of $I_{KGE}$ from 0.62 (HBV) and
0.61 (TOPMODEL) to 0.65. In 12% of the MOPEX catchments, the HSC module outperforms HBV, and in
not a single catchment did the calibrated HBV outperform the HSC. Comparing with TOPMODEL, the HSC
outperformed in 8% of the catchments, and in only 1% of catchments TOPMODEL has a better
performance. Not surprisingly, the $I_{KGE}$ of HSC-MCT model was slightly reduced to 0.59, due to the non-
calibrated $S_{uMax}$, but still comparably well performed as HBV (0.62) and TOPMODEL (0.61). This illustrates
the robustness of both the HSC approach to derive the spatial distribution of the root zone storage
capacity (β) and the efficiency of the MCT method to estimate the root zone storage capacity ($S_{uMax}$).
Moreover, the new module allows us to map out the saturated area, which has potential capability to be
used for broader hydrological, ecological, climatological, geomorphological, and biogeochemical studies.

**Acknowledgement:**
This study was supported by National Key R&D Program of China (2017YFE0100700).

**Author contributions:**
H.G. and H.H.G.S. designed research; H.G. performed research; C.B., C.S., D.T and H.G. provided data,
among which the dynamics of the saturation areas data in the BB was provided by C.B. C.S., and D.T.; H.G.
analysed data; C.B. was involved in the interpretation of some of the modelling work in the BB; H.G. M.H,
and H.H.G.S. wrote the paper; CS and DT extensively edited the paper, and provided substantial comments
and constructive suggestions for scientific clarification.



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

Table 1. The parameters of the models, and their prior ranges for calibration. ($^*S_{uMax}$ is a parameter in HBV,
TOPMODEL and the HSC model, but HSC-MCT model does not have $S_{uMax}$ as a free parameter; $^{**}$ β is a parameter in
HBV model, but not in TOPMODEL, HSC and HSC-MCT models)

| Parameter | Explanation | Prior range for calibration |
|---|---|---|
| $S_{iMax}$ (mm) | Maximum interception capacity | 2 |
| $S_{uMax}$ (mm) $^*$ | The root zone storage capacity | (10, 1000) |
| β (-)$^{**}$ | The shape of the storage capacity curve | (0.01, 5) |
| $C_e$ (-) | Soil moisture threshold for reduction of evaporation | (0.1, 1) |
| $D$ (-) | Splitter to fast and slow response reservoirs | (0, 1) |
| $T_{lagF}$ (d) | Lag time from rainfall to peak flow | (0, 10) |
| $K_f$ (d) | The fast recession coefficient | (1, 20) |
| $K_s$ (d) | The slow recession coefficient | (20, 400) |



Table 2. The water balance and constitutive equations used in models. (Function (15)$^*$ is used in the HBV model, but
not used in the TOPMODEL, HSC and HSC-MCT models)

| reservoirs | Water balance equations | Constitutive equations |
|---|---|---|



| Interception reservoir | $\dfrac{\mathrm{d}\,S_i}{\mathrm{d}\,t} = P - E_i - P_e$ (8) | $E_i = \begin{cases} E_p; S_i > 0 \\ 0; S_i = 0 \end{cases}$ (9) |
| | | $P_e = \begin{cases} 0; & S_i < S_{iMax} \\ P; & S_i = S_{iMax} \end{cases}$ (10) |
| Unsaturated reservoir | $\dfrac{\mathrm{d}\,S_u}{\mathrm{d}\,t} = P_e - E_a - R_u$ (11) | $\dfrac{R_u}{P_e} = \left(\dfrac{S_u}{S_{uMax}}\right)^{\beta}$ (12)* |
| | | $\dfrac{E_a}{E_p - E_i} = \dfrac{S_u}{C_e S_{uMax}}$ (13) |
| Splitter and Lag function | | $R_f = R_u D$ (17); $R_s = R_u(1-D)$ (14) |
| | | $R_{fl}(t) = \sum\limits_{i=1}^{T_{lagf}} c_f(i) \cdot R_f(t-i+1)$ (15) |
| | | $c_f(i) = i / \sum\limits_{u=1}^{T_{lagf}} u$ (16) |
| Fast reservoir | $\dfrac{\mathrm{d}\,S_f}{\mathrm{d}\,t} = R_f - Q_f$ (17) | $Q_f = S_f / K_f$ (18) |
| Slow reservoir | $\dfrac{\mathrm{d}\,S_s}{\mathrm{d}\,t} = R_s - Q_s$ (19) | $Q_s = S_s / K_s$ (20) |







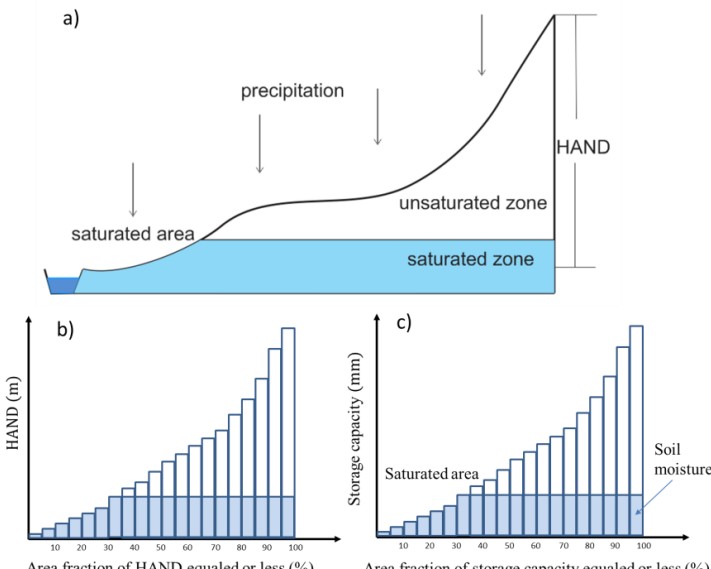


Figure 1. The perceptual model of the HAND-based Storage Capacity curve (HSC) model. a) shows the representative

hillslope profile in nature, and the saturated area, unsaturated zone and saturated zone; b) shows the relationship

between HAND bands and their corresponded area fraction; c) shows the relationship between storage capacity-

area fraction-soil moisture-saturated area, based on the assumption that storage capacity linearly increases with

HAND values.





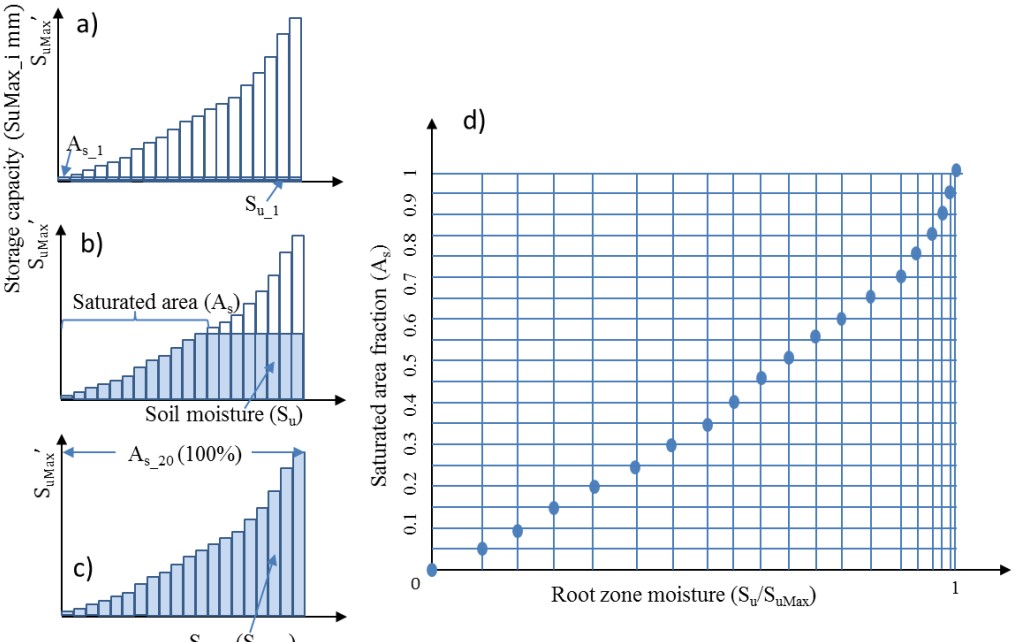


Figure 2. The conceptual model of the HSC model. a), b) and c) illustrate the relationship between soil moisture ($S_u$)
and saturated area ($A_s$) in different soil moisture conditions. In d), 20 different $S_u$-$A_s$ conditions are plotted, which
allow us to estimate $A_s$ from $S_u$.

```
┌──────────────┐
│  DEM map     │
└──────────────┘
       │
┌──────────────┐
│  HAND map    │
└──────────────┘
       │
┌──────────────────┐
│ HAND – area fraction │
└──────────────────┘
       │         ┌─────────────────────────────────┐
       │         │ Assumption: storage capacity linearly │
       │         │ increases with HAND              │
       │         └─────────────────────────────────┘
┌─────────────────────────────┐
│ storage capacity – area fraction │
└─────────────────────────────┘
       │
┌──────────────────────────────────────────┐
│ catchment root zone moisture – saturated area │
└──────────────────────────────────────────┘
       │         ┌─────────────────────────────────┐
       │         │ Assumption: saturation excess flow is the │
       │         │ dominant runoff generation mechanism │
       │         └─────────────────────────────────┘
┌──────────────────────────────────────────────────┐
│ catchment root zone moisture – runoff generation area │
└──────────────────────────────────────────────────┘
```


Figure 3. The procedures estimating runoff generation by the HSC model and its two hypotheses.




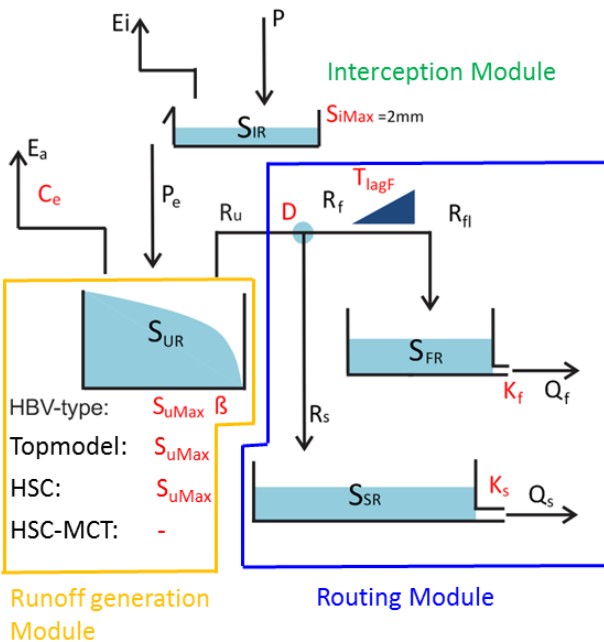


Figure 4. Model structure and free parameters, involving four runoff generation models (HBV-type, TOPMODEL, HSC, and HSC -MCT). HBV-type has $S_{uMax}$ and beta two free parameters; TOPMODEL and HSC models have $S_{uMax}$ as one free parameter; and HSC-MCT model does not have free parameter. In order to simplify calibration process and make fair comparison, the interception storage capacity ($S_{iMax}$) was fixed as 2mm.






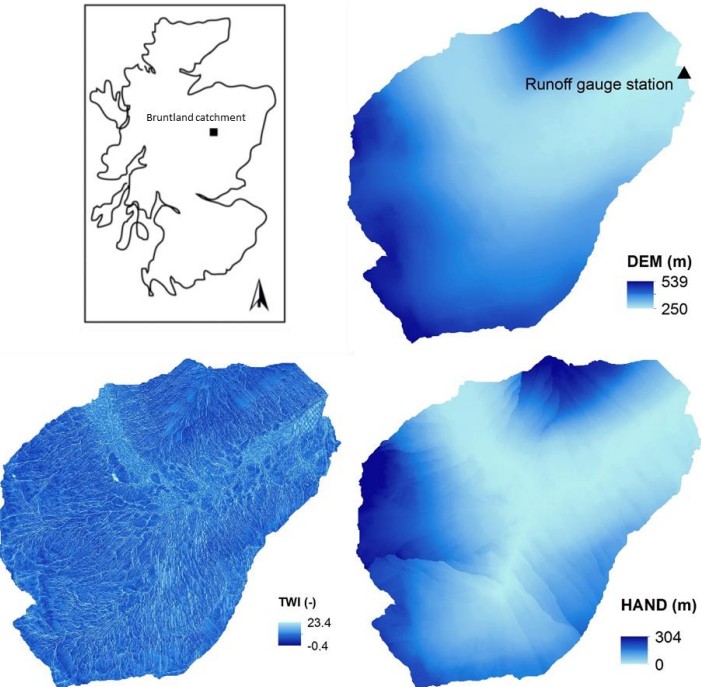


Figure 5. (a) Study site location of the Bruntland Burn catchment within Scotland; (b) digital elevation model (DEM)
of the Bruntland Burn catchment; (c) the topographic wetness index map of the Bruntland Burn catchment; (d) the
height above the nearest drainage (HAND) map of the Bruntland Burn catchment.





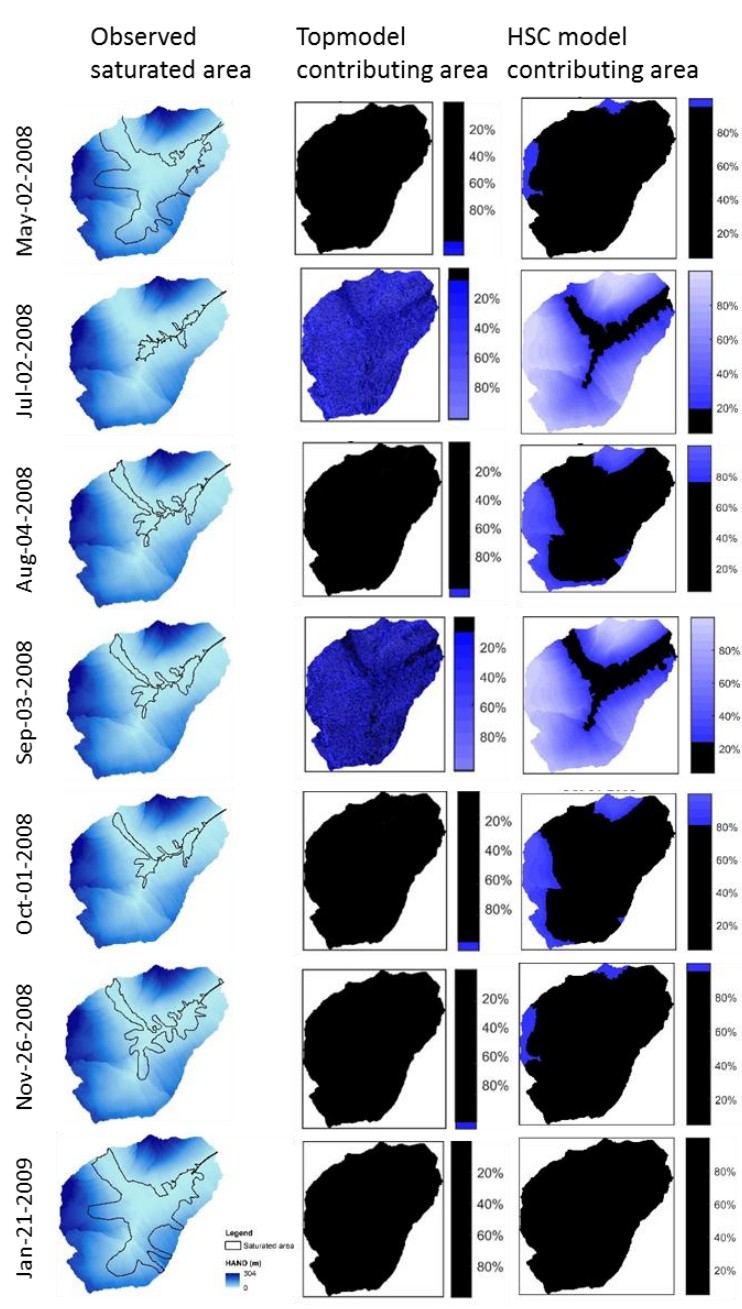


Figure 6. The measured saturated areas and the simulated contributing areas by TOPMODEL and HSC models.




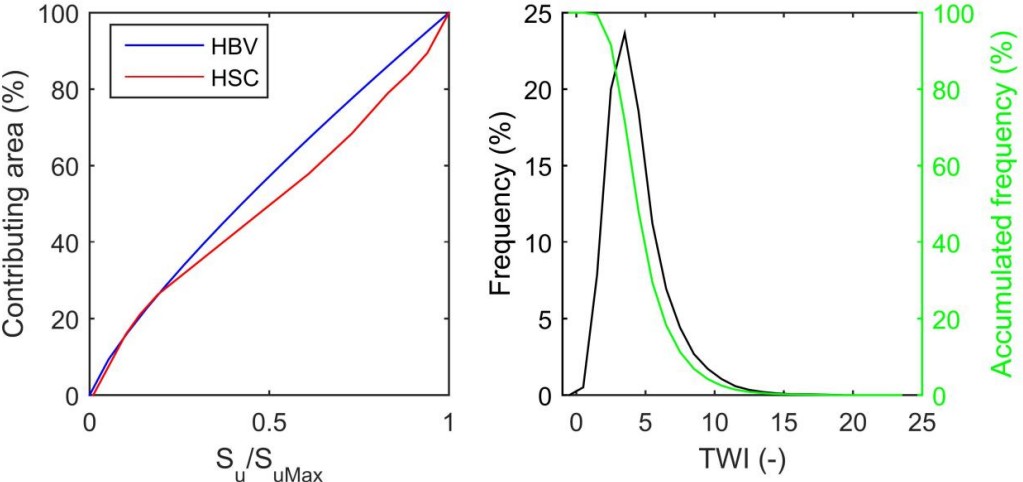


Figure 7. The curves of the beta function of HBV model, and the $S_u$-$A_s$ curve generated by HSC model (the left figure).
The frequency and accumulated frequency of the TWI in the Bruntland Burn catchment (the right figure).

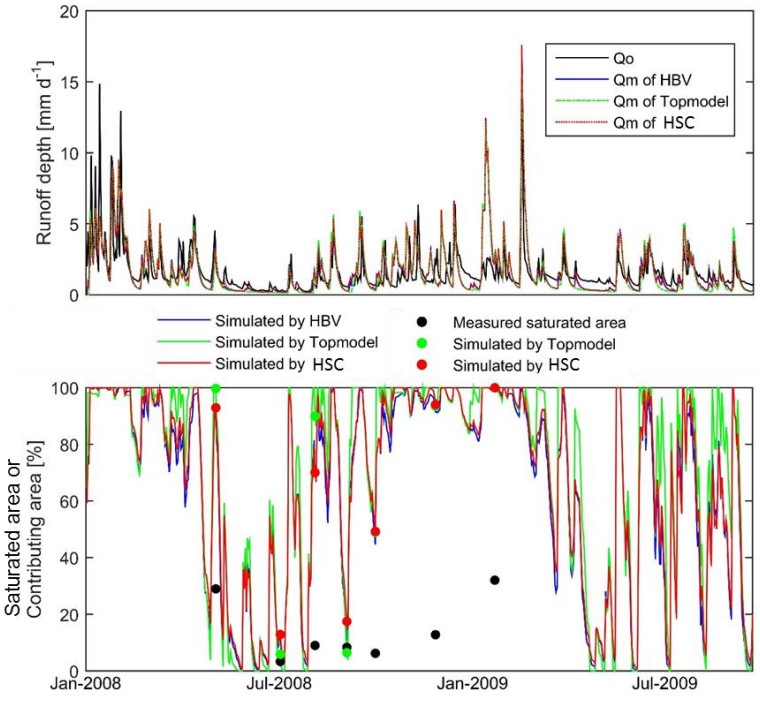




Figure 8. The observed hydrograph (Qo, black line) of the Bruntland Burn catchment in 2008. And the simulated
hydrographs (Qm) by HBV model (blue line), TOPMODEL (green dash line), HSC model (red dash line). And the
comparison of the simulated relative soil moistures, i.e. HBV (blue line), TOPMODEL (green line), HSC (red line). And
the observed saturated area of 7 days (black dots), and the correspondent simulated contributing area by
TOPMODEL (green dots) and by HSC model (red dots).


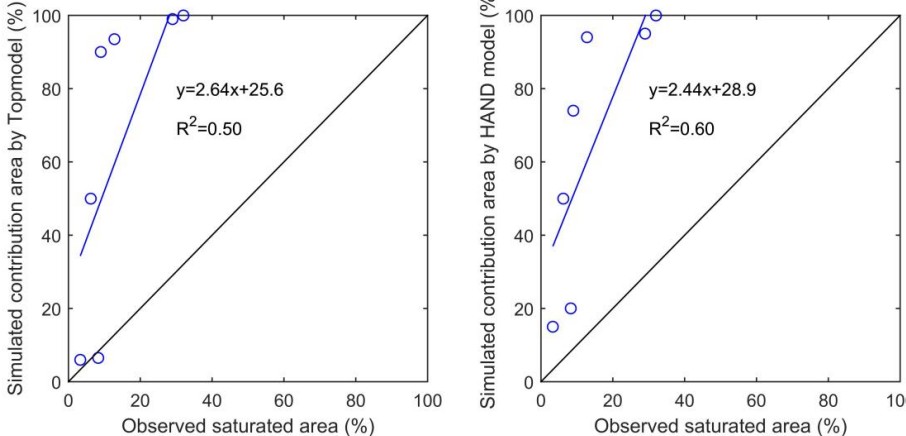


Figure 9. The comparison of the observed saturated area and simulated contributing areas by TOPMODEL and HSC
models.



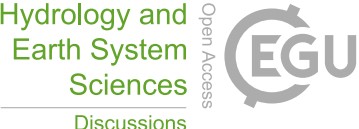



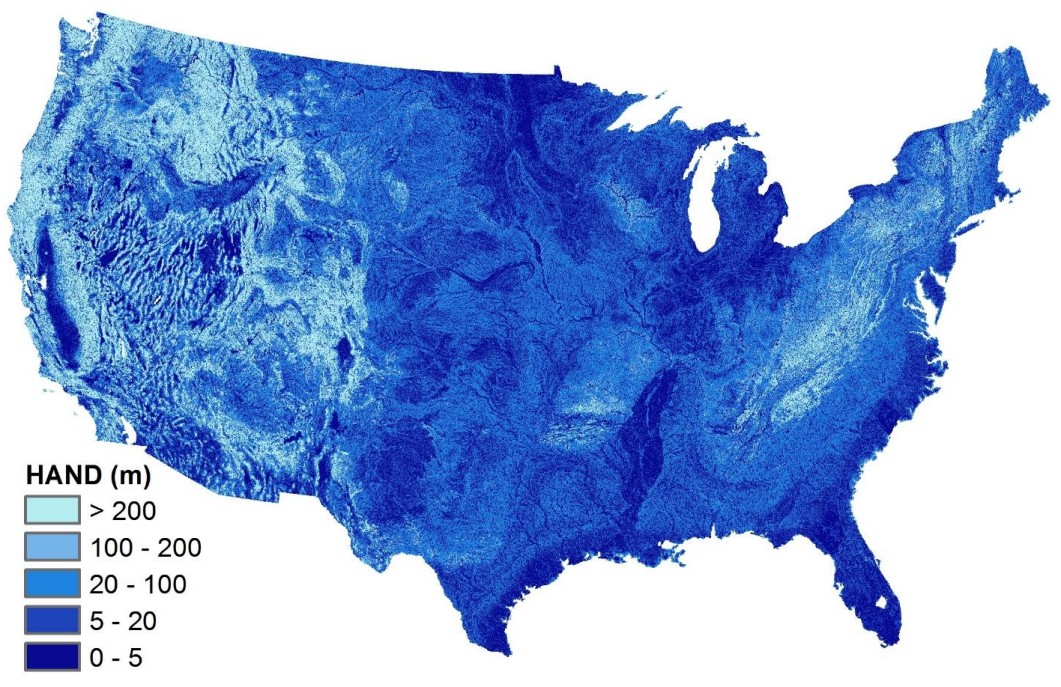


Figure 10. The Height Above the Nearest Drainage (HAND) map of the CONUS.


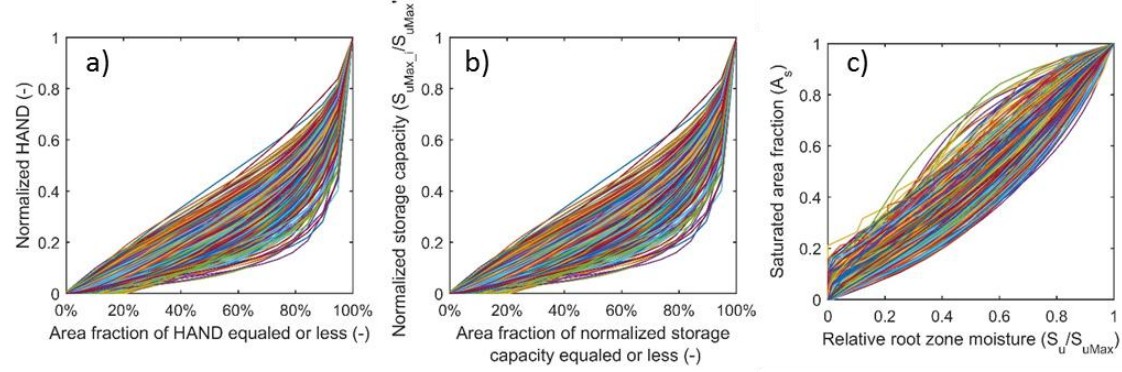


Figure 11. a) The profiles of the normalized HAND of the 323 MOPEX catchments; b) the relations between area
fraction and the normalized storage capacity profile of the 323 MOPEX catchments; c) the $S_u$-$A_s$ curves of the HSC
model which can be applied to estimate runoff generation from relative soil moisture for the 323 MOPEX catchment.




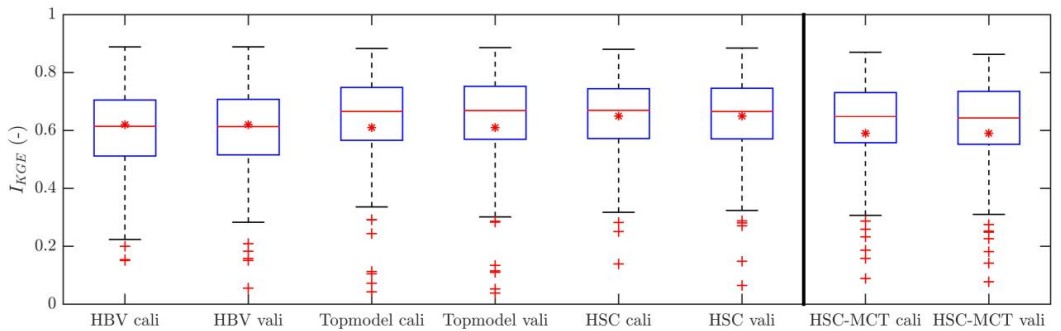


Figure 12. The comparison between the HBV, the TOPMODEL, the HSC, and the HSC-MCT models


Figure 13. Performance comparison of the HSC and HSC-MCT models compared to two benchmarks models: HBV
and TOPMODEL, for the 323 MOPEX catchments.