# Peer review of "A simple topography-driven and calibration-free runoff generation module"

_Hydrology and Earth System Sciences, 2018_

## Referee Comment (RC1) · Anonymous Referee #1 · 15 Apr 2018

This paper proposes a new method to estimate the spatial distribution of soil water storage capacity in catchments based on DEM and long-term water balance data. The shape parameter of the distribution is estimated based on HAND values derived from DEM. The average storage capacity over a catchment is estimated, by the MCT method, from climatological and vegetation data (i.e., long-term precipitation, runoff and seasonal NDVI). This method is evaluated in an experimental catchment and MOPEX catchments. This paper provides a novel method to estimate the spatial distribution of soil water storage capacity, which is an important feature for catchment hydrology but is usually not available. I have some comments for the authors to consider before the publication.

Comments:

1. As shown in Figure 2, saturated area and runoff coefficient is dependent on the initial soil moisture condition ($\frac{S_u}{S_{uMax}}$). However, within a time step (daily in this paper), rainfall in the early time causes the increases of $\frac{S_u}{S_{uMax}}$ and runoff coefficient. Therefore, the runoff coefficient during a day is also affected by rainfall, and runoff coefficient is a function of $\frac{S_u}{S_{uMax}}$, $\beta$, and $\frac{S_{uMax}}{P_e}$ (Moore, 1985; Wang, 2018). Given the values of $\frac{S_u}{S_{uMax}}$ and $\beta$, equation 12 in Table 2 underestimates $\frac{R_u}{P_e}$ in a day, and this underestimation increases with increasing rainfall depth. Similarly for HSC module, if $A_s$ is determined by $\frac{S_u}{S_{uMax}}$ at the beginning of a day, the effect of $P_e$ on runoff coefficient is not considered.

2. For HSC and HSC-MCT modules: the relationship $A_s = f(S_u/S_{uMax})$ is obtained from the HAND values (Figure 2d). This relationship can be obtained by fitting a distribution function to the CDF of normalized HAND values (e.g., Figure 2b). For example, if the distribution used in Xinanjiang (Zhao, 1992) and VIC (Wood et al., 1992) is applied, the CDF of storage capacity is:

$$F(C) = 1 - \left(1 - \frac{C}{C_m}\right)^{\beta} \tag{1}$$

The mean value of storage capacity is:

$$S_{uMax} = \frac{C_m}{\beta + 1} \tag{2}$$

Substituting $C_m$ from equation (2) into equation (1), we obtain

$$F(C) = 1 - \left(1 - \frac{1}{\beta + 1}\frac{C}{S_{uMax}}\right)^{\beta} \tag{3}$$

$\frac{C}{S_{uMax}}$ is the normalized HAND values (HAND values divided by its average) in Figure 2b. The value of $\beta$ is estimated by fitting the distribution. Then the relationship $A_s = f(S_u/S_{uMax})$ is obtained:

$$A_S = 1 - \left(1 - \frac{S_u}{S_{uMax}}\right)^{\frac{\beta}{\beta+1}} \tag{4}$$

The curve for $A_S \sim \frac{S_u}{S_{uMax}}$ is concave since $\frac{\beta}{\beta+1} < 1$. $A_S \sim \frac{S_u}{S_{uMax}}$ for HBV is concave or convex. The distribution corresponding to the SCS curve number method is another alternative (Wang, 2018), and the CDF is written as:

$$F(C) = 1 - \frac{1}{a} + \frac{\frac{C}{S_{uMax}}+1-a}{a\sqrt{\left(1+\frac{C}{S_{uMax}}\right)^2 - 2a\frac{C}{S_{uMax}}}} \tag{5}$$

Where $a$ is the shape parameter. The $A_S \sim \frac{S_u}{S_{uMax}}$ relation for equation (5) is:

$$A_S = 1 - \frac{1}{a} + \frac{\frac{C}{S_{uMax}}+1-a}{a\sqrt{\left(1+\frac{C}{S_{uMax}}\right)^2 - 2a\frac{C}{S_{uMax}}}} \tag{6}$$

Where $\frac{C}{S_{uMax}} = \frac{\left(2-a\frac{S_u}{S_{uMax}}\right)\frac{S_u}{S_{uMax}}}{2\left(1-\frac{S_u}{S_{uMax}}\right)}$. It is interesting to test the goodness-of-fit of these two distributions (equations 3 and 5) to the empirical $A_S \sim \frac{S_u}{S_{uMax}}$ from the HAND-based values.

3. Line 167: I guess that "SEF" represents "saturation excess flow". But spell out "SEF" which is not defined before. Same for SOF and SFF.
4. It may be good to add a map to show the spatial distribution of $S_{uMax}$.
5. For figures with subplots, it is better to add "(a)" and "(b)", e.g., Figure 7 and Figure 8.
6. In the comparison of Figure 7a, is it possible to convert the distribution of TWI information to $A_S \sim \frac{S_u}{S_{uMax}}$ and add it to Figure 7a?
7. In Figure S4 (catchment 08171000), the calibrated $\beta$ for HBV is about 1.8. $\beta = 1.35$ is very close to the curve of HSC. But it seems that the performance of HSC is better than HBV (lines 500-501 and Figure S3). Since $\beta$ of HBV is calibrated, why is the calibrated value of $\beta$ not around 1.35? Is this due to the effect of other calibrated parameters? Discussion on this may be helpful.
8. This comment is related to the previous one. For the comparison between HSC and HSC-MCT, is the calibrated $S_{uMax}$ for HSC similar to the estimated $S_{uMax}$ by MCT method? How about the other calibrated parameters (e.g., $D$, $K_f$, $K_s$, $T_{lagF}$) among the models (HBV, TOPMODEL HSC, and HSC-MCT)?

References

Moore, R. J. (1985), The probability-distributed principle and runoff production at point and basin scales, *Hydrol. Sci. J.*, 30, 273-297.
Wang, D.: A new probability density function for spatial distribution of soil water storage capacity leads to SCS curve number method, *Hydrol. Earth Syst. Sci. Discuss.*, https://doi.org/10.5194/hess-2018-32, in review, 2018.

Wood, E. F., D. P. Lettenmaier, and V. G. Zartarian (1992), A land‑surface hydrology parameterization with subgrid variability for general circulation models, *J. Geophys. Res., Atmospheres*, 97(D3), 2717-2728.

Zhao, R. (1992), The Xinanjiang model applied in China, *J. Hydrol.*, 135, 371-381.

---

## Referee Comment (RC2) · Anonymous Referee #2 · 28 May 2018

This study proposed the HAND-based storage capacity curve (HSC) for runoff generation parameterization in hydrological models. I like the idea to provide parameter-reduced modules for hydrological modeling, considering the significant uncertainty in parameter calibration. However, the benefits for reducing parameter uncertainty by the HSC module were not illustrated by the current results. The authors claimed that the HSC/HSC-MCT module possess higher robustness and bear the potential to be implemented in prediction of ungauged basins, which however are not convincing from the current results. The benefits of the HSC/HSC-MCT modules need to be further discussed. First, the HSC module obviously overestimated the saturated area fraction in the Bruntland Burn (BB) basin. The improvement for the correlation from 0.5 (TOP-MODEL) to 0.6 (HSC) is rather small, which does not make any sense for illustrating

the reduced deviations between the observed and simulated saturated area fractions. The model performance in validation period were not evaluated in the BB basin. Although the HSC module gained performance improvements on high flows in both calibration and validation periods in many MOPEX basins, the performance gains on low flows were not investigated. Moreover, the non-parameter HSC-MCT module produced much lower performance than the HSC module in many MOPEX cases. The gains for the HSC module should be attributed to the parameter calibration procedure, and potentially demonstrated the failure of the MCT module for many MOPEX cases. Second, it is not fair to call the proposed modules are calibration-free. The HSC module also implied two parameters for the model application. The stream initiation threshold area was not included for the model calibration, but was tested to calculate the HAND values. I guess this threshold area should be tested in calculation experiments to prepare the model results using the HSC modules. The effects of this parameter on the performance of the HSC modules were not investigated in the results. Including this parameter in the calibration procedure would most likely to improve the model performance. Moreover, model calibration procedures are required to determine the remaining parameter values in Table 1 for both the applications of HSC and HSC-MCT modules. The benefits to reduce parameter uncertainty by excluding one-two parameters by the HSC and HSC-MCT modules were not clear. Considering the preparation of HAND values using DEM dataset in the HSC and HSC-MCT modules, the computational cost should be much higher than the calibrated modules in HBV and TOPMODEL. Some other major concerns on the results are listed as follow. 1. Figure 6, why not show the saturated area fraction simulated by the HBV module? Is the HBV module spatially discretized for the model application? 2. Figure 7, why was the beta value of 0.98 used for the HBV module? Was this beta value derived from the model calibration? Have you tried other beta values for the comparison between HBV and HSC modules? Why not show this curve for the TOPMODEL module? What is your purpose to show the frequency of TWI? Could you also show the frequency of HAND? 3. Figure 8, the label for soil moisture was missed. It is very difficult to find the observed soil moisture

(or you don't have?). Can you label (a-b) for the two subplots? For the second and fourth events, the TOPMODEL matched the observed saturated area fractions very well. How to explain this? Please refine the caption, which is very difficult to understand. 4. Figure 9, as I suggested before, a correlation coefficient does not make any sense to illustrate the deviations between the observed and simulated saturated area fractions, given that only seven observation events. Could you use some other metrics to compare the bias or deviation errors between the observed and simulated saturated area fractions? 5. Figures 10-11, what do you intend to say from these figures? 6. Figure 12, I suggest to compare the values for IKGL as well. The models were calibrated on both IKGL and IKGE, there should be strong trade-off between these two objective functions. That means the HSC module possibly sacrificed the performance for low flows (IKGL) to improve the performance for high flows (IKGE). The evaluated modules mainly differed on the calculation of soil storage capacity, which has significant effect on the generation of low flows. In my opinion, performance for low flows should also be an important indicator for the validity of the runoff generation assumptions. Minor concerns: 1. I suggest to remove "Calibration-free" from the title. HSC module needs to be calibrated, and the HSC-MCT module performed poorly in many MOPEX cases. 2. Lines 30-33, I am not convinced to agree with this from the current results. What do you mean "facilitated effective visualization of the saturated area"? Is it important? 3. Introduction is too long from my taste. It is very difficult to get the motivations of this study from this section. I would suggest to refine it. 4. Lines 213-214, remove "Hydrological. . . inevitable". Line 217, what do you mean "HAND contours are parallel in runoff generation"? Is that possible derived from the DEM? 5. Line 227, could you please add more details for the calculation of HAND values? 6. Line 314, how did you define the pareto-frontier? Did you use the Euclidean distance or threshold values? 7. Section 3.1, could you please add some details on the climatic and hydrological data in the BB basin? Any ground gauged stations do you have there? 8. Also in section 3.1, could you introduce the spatial interpolation of the field mapping of the saturated areas? 9. Line 369-374, move to the methodology section. 10. Lines 415-420, move

to the methodology section. 11. Lines 455-456, 'dramatically improved' may be not fair. 'simultaneously maintaining model robustness and consistency' is also not convinced by the results. 12. Lines 491-496, it is not fair to only discuss the cases where HSC/HSC-MCT outperformed the benchmark modules. Why not discuss the reasons for the cases where HSC/HSC-MCT produced lower performance? 13. Discussion is also too much. It is difficult to get the main messages from the long text. Maybe remove lines 508-522, and lines 539-556. 14. Lines 646-647, maybe it is not so important to say as one of the conclusions here. 15. There are many sentence started with 'And', this is very strange (kind of grammatical error).

---

## Referee Comment (RC3) · Anonymous Referee #3 · 30 May 2018

General comments

This study presents a new concept for runoff generation description in conceptual hydrologic models. The new approach is based on HAND (height above nearest drainage) information derived from digital elevation model. The methodology is tested for two cases: (1) small experimental catchment in Scotland; (b) MOPEX dataset in the US. Results are compared against observed saturation patterns (in case 1) and discharge observations (both cases), as well against simulations of two other conceptual hydrologic models. The authors conclude that the new concept compares well with other two calibrated models and allows to describe spatial distribution of the root zone storage capacity.

Overall the topic is interesting and within the scope of HESS. However I fully agree

with referee #2 that manuscript will benefit from some strengthening of the take home message, i.e. by providing more thorough and additional process based evaluation of results. I missed some more thorough process based interpretation of the reasons for similarity/differences in saturated area patterns for case 1 (catchment in Scotland). It seems to me that the differences between observed and simulated saturated area patterns are quite large and does not support well the interpretation that the new concept is better than the other approaches (yes, it is a little bit better than the models but for some days quite far from the observations and not convincing well the benefits of the proposed approach). The results for case 2 (MOPEX dataset) present mostly a statistical comparison of efficiency numbers (average, median), but does not tell much about the seasonal, geological, vegetation, climate and flow characteristics impacts on the efficiency evaluation. Some classification of catchments according e.g. similar TWI or HAND based indices, runoff regime indices, etc. and subsequent separate analysis of results for such groups will allow to more clearly indicate the role of different physiographic conditions on the results. I'm not sure to what extent can be the presented example for one catchment generalised for the other catchments, so some more assessment will be useful here. For example the results indicate that the new concept is better for mild sloped catchments, so a figure showing the results for all such catchments compared to the others will be interesting. Along the same line, similar evaluation for different geological/vegetation/climate groups of catchments with some process based interpretation of results will shed more light about what new and different information is obtained in the new HAND based storage capacity estimates compared to TWI (research question 2). (I'm missing a clear answer here - the maps are quite difficult to read, particularly for people which are not experts on the local situation). The discussion of the results is in some parts too vague and not linked well with the results (e.g. section 6.2). On the other hand there are much more MOPEX based studies and some of them indicate better model performance (e.g. for HBV model, e.g. Kollat et al, 2012, https://doi.org/10.1029/2011WR011534) than found here. So, some more thorough link with existing MOPEX studies will be thus suggested.

Specific comments

1) Abstract: Please consider to be more specific about how much better the HSC concept is in reproducing the spatio-temporal pattern of the observed saturation areas, as well as in comparison with calibration and validation efficiencies of other conceptual models.

2) Figure 1 and associated text. I wonder to what extent the new concept (HAND is proportional to storage capacity) reflects different geomorphological and geological processes? In which geological conditions one can apply the concept?

3) Figure 6. The colour legends are very confusing. It will be easier to have the same legend for all maps.

4) It will be interesting to provide, as a supplement, a list of used catchments with the results.

---

## Author Comment (AC1) · 7 Jun 2018

We thank the Anonymous Reviewer 1 for recognising the innovation and the importance of this paper. We also appreciate all his/her constructive comments, which are valuable to improve the quality of this manuscript. For the detailed comments, please find our responses in below.

Replies:

1. The influence of $S_{uMax}/Pe$ on runoff coefficient estimation (Moore, 1985; Wang, 2018) will be discussed in the revised paper.

2. In the manuscript, we have compared the model performance of HSC and HSC-MCT with HBV and TOPMODEL (as benchmarks), and found that the HSC module

performed better in both calibration and validation. HBV is a good benchmark, be-cause it has a relatively straightforward way of representing the runoff threshold in the root zone, albeit by calibration. TOPMODEL is also a good benchmark, because it uses a topographical index to define the runoff threshold. In our approach, the spatial distribution of the HAND values is used to derive the spatial distribution of the runoff (connectivity) thresholds, but from another topographical perspective than TOPMODEL. We agree that it would be interesting to test the goodness-of-fit of the Cumulative Distribution Function (CDF) of HSC with not only the HBV, but also the Xinanjiang, GR4J and SCS. However it might be worthwhile to clarify that the intention of is to propose a new runoff generation module (HSC), which is, to some extent, supported by large-sample ecological field observation, and free of calibration, rather than comparing the CDF of HSC with other existing modules.

3. The full names of the SEF, and SOF will be clearly defined.

4. The $S_{uMax}$ for each MOPEX catchment in the HSC-MCT module was obtained in our previous study (Gao et al., 2014). We used the amount of root zone storage capacity, which ecosystems need to overcome drought periods (dry spells) with 20 years return period ($S_{R20y}$), as a proxy for $S_{uMax}$. The details of the method to derive the $S_{R20y}$ can be found in Gao et al., 2014.

5. Figure 7 and 8 will be revised.

6. It is a good suggestion to put the TOPMODEL and HBV curves together, and compare their shape. But it is a difficult task, due to the different model assumption and concept. And to our best knowledge, we haven't found similar studies that systematically compare the TOPMODEL curves with HBV curves, which might indicate that this is not an easy task to be perform within a short time. Furthermore, in this study we just wanted to compare the model performance of HSC with HBV and TOPMODEL, rather than to unify all model approaches.

7. The effect of other calibrated parameters on model calibration and efficiency will be

discussed.

8. The comparison of the calibrated $S_{uMax}$ and the estimated $S_{uMax}$ by MCT can be found in Gao et al., 2014. For the other calibrated parameters, their effect on model performance will be discussed in the revised manuscript. It is worth noting that all models use the same model structure and prior range of remaining parameters (i.e. interception and response modules) to exclude the impact of other processes, and guarantee that the comparison of runoff generation modules is fair.

References:

Moore, R. J. (1985), The probability-distributed principle and runoff production at point and basin scales, Hydrol. Sci. J., 30, 273-297.

Wang, D.: A new probability density function for spatial distribution of soil water storage capacity leads to SCS curve number method, Hydrol. Earth Syst. Sci. Discuss., https://doi.org/10.5194/hess-2018-32, in review, 2018.

Gao H, Hrachowitz M, Schymanski SJ, Fenicia F, Sriwongsitanon N, Savenije HHG. 2014. Climate controls how ecosystems size the root zone storage capacity at catchment scale. Geophysical Research Letters 41 (22): 7916–7

---

## Author Comment (AC2) · 7 Jun 2018

We thank Anonymous Referee 2 for the very constructive and detailed comments. Here are our replies.

**Overestimation of the saturated area.** The overestimation of the saturated area is most likely caused by the different definition of saturated areas in field measurement and in hydrological models. The discussion and interpretation of the overestimation of the saturated area fraction in the BB basin are described in Line 604-614.

**Model validation in BB basin.** We will add the model performance in the validation period, and evaluate the models in BB.

**Model performance on low flow.** The performance gains on low flow ($I_{KGL}$) have

been investigated and shown in Figure RC2.

**Calibration-free.** We may politely insist that the HSC-MCT is a calibration-free runoff generation module. We agree that the threshold area for stream initiation is important while generating HAND maps. But the threshold area can be determined based on observation rather than calibration, although the threshold area varies in different climate, geology and landscape classes. The limitation of the fixed threshold area has been discussed in Line 601-604.

**MCT method.** MCT is an approach to estimate the $S_{uMax}$ by measurable input. But since we fixed this parameter as $S_{R20y}$ (the amount of root zone storage capacity, which ecosystems need to bridge droughts with 20 years return period), which may also vary in different ecosystems. Improving the MCT to allow more flexible estimation for different ecosystems will be promising to improve model performance, which is discussed in Line 595-599.

**Computational cost.** The discussion on the computational cost will be added in the revised manuscript.

More concerns:

1. The saturated area fraction simulated by HBV is presented in Figure 8b. But the HBV cannot explicitly generate the spatial discretization of saturation area.

2. Yes, the beta value of 0.98 is the averaged calibrated value of beta. Please note that the intention of the HSC module is to propose a new runoff generation module, which is, to some extent, supported by large-sample ecological field observation, and free of calibration, rather than fitting the CDF of HSC with other existing curves/modules. The purpose to show the TWI frequency of TOPMODEL is to demonstrate the curve that we used to estimate runoff in TOPMODEL. The HSC curve in Figure 7 is derived from the spatial distribution of HAND, therefore the HAND distribution curve is not shown in this figure.

[Figure]

3. Label a-b will be added in Figure 8, and the caption will be refined. The averaged relative soil moisture of root zone ($Su/S_{uMax}$) at catchment scale is used to estimate the proportion of saturated area ($As$). In the manuscript, we demonstrated the estimated As rather than the soil moisture, because As is more directly linked with runoff generation simulation. Yes, TOPMODEL does perform better in the second and the fourth events, but generally HSC performs better than TOPMODEL (evaluated by $R^2$ and $I_{KGE}$).

4. This is a good suggestion. IKGE might be a better metric to evaluate model performance on saturated area fraction estimation. Evaluated by IKGE, HSC also performs better than the TOPMODEL, although both HSC and TOPMODEL do not perform well (-3.0 for HSC, and -3.4 for TOPMODEL). The reasons for the unsatisfactory results are discussed in Line 604-614.

5. We intended to present the procedures to derive the HSC curves for the MOPEX catchment, which are helpful for readers to understand how the HSC module works.

6. The results of $I_{KGL}$ will be presented.

Minor concerns:

1. We may politely insist that the HSC-MCT module is calibration-free and performs equally well or better as a calibrated model. There are two reasons. Firstly, as we clarified in the above, HSC is directly derived from the HAND distribution in a DEM, without any calibration. Secondly, HSC-MCT performs comparably well with HBV. Since the median IKGE value of HSC-MCT is 0.65, which is a better performance compared to HBV (0.61). And the averaged IKGE value of HSC-MCT is 0.59, which is comparable to 0.62 (HBV). So it is fair to say the model performance of the calibration-free HSC-MCT and HBV are comparable.

2. This will be rephrased.

3. We will refine introduction.

4. This will be rephrased.

5. More detailed about the calculation of HAND can be found in Rennó et al., 2008; Gharari et al., 2011.

6. Please refer to Vrugt et al., 2003.

7. Yes, we will add more details on the climatic and hydrological data in the BB catchment.

8. The saturation maps are not interpolated. They are generated directly by field mapping, and a global positioning system (GPS) was used to delineate the boundary of saturation areas (Ali et al., 2014).

9. We will revise it as suggested.

10. We will revise it as suggested.

11. This will be refined.

12. The reasons for the cases where HSC/HSC-MCT produced lower performance will be discussed.

13. The discussion will be revised.

14. This will be revised.

15. This will be revised.

References:

Renno, C. D., Nobre, A. D., Cuartas, L. A., Soares, J. V., Hodnett, M. G., Tomasella, J., and Waterloo, M. J.: HAND, a new terrain descriptor using SRTM-DEM: Mapping terra-firme rainforest environments in Amazonia, Remote Sens. Environ., 112, 3469–3481, doi:10.1016/j.rse.2008.03.018, 2008.

Gharari, S., Hrachowitz, M., Fenicia, F., and Savenije, H. H. G.: Hydrological landscape

classification: investigating the performance of HAND based landscape classifications in a central European meso-scale catchment, Hydrol. Earth Syst. Sci., 15, 275–3291, doi:10.5194/hess-15-3275-2011, 2011

Vrugt, J. A., Gupta, H. V., Bastidas, L. A., Bouten, W., and Sorooshian, S.: Effective and efficient algorithm for multiobjective optimization of hydrologic models, Water Resour. Res., 39, 1214, doi:10.1029/2002wr001746, 2003.

Ali, G., Christian Birkel, Doerthe Tetzlaff, et al. A comparison of wetness indices for the prediction of observed connected saturated areas under contrasting conditions[J]. Earth Surface Processes and Landforms, 2014, 39(3):399-413.

[Figure]

**Fig. 1.** Figure RC2. Model performance on low flow (IKGL).

---

## Author Comment (AC3) · 7 Jun 2018

We thank Anonymous Referee 3 for all his/her constructive comments and useful suggestions. Here are our replies to the comments:

1. The overestimation of the saturated area is most likely caused by the different definition of saturated area in field measurement and in hydrological models. The discussion and interpretation of the overestimation of the saturated area fraction in the BB basin are described in Line 604-614.

2. It is worthwhile to test the impact of seasonal, geological, vegetation, climate and flow characteristics on model efficiency. Actually, we have conducted a study with the MOPEX data to test the impact of vegetation, climate, geology, topography, and other

catchment characteristics on the shape of the beta function, and found that the topographic information has the most significant impact on the shape of beta function (Gao et al., 2018). In this study we also found the impact of other characteristics on model efficiency not as clear as topography. And the new concept module (HSC) has better performance in mildly sloping catchments, which means topography impacts on the efficiency of HSC module. Therefore, we merely discussed the impact of topography on model efficiency in the manuscript.

3. The discussion will be revised to be better linked with the results.

4. We compared the HBV model performance in MOPEX catchments with other studies (e.g. Ye et al., 2014). We will also refer to Kollat et al. (2012) in the revised manuscript.

Specific comments:

1. We will rephrase the abstract.

2. As has been discussed in Section 6.2, topography, with fractal characteristic, is often the dominant driver of runoff, as well as being a good integrated indicator for vegetation cover, rooting depth, root zone evaporation and transpiration deficits, soil properties, and even geology. But quantifying to what extent the HSC concept reflects different geomorphological and geological processes is still a challenge (Rempe and Dietrich, 2014; Gomes, 2016), which needs further investigation. This limitation will be further discussed in the revised manuscript.

3. I will redo the plotting of Figure 6, to make sure they have the same legend for all maps.

4. Yes, a list of the used catchments will be added as SI material.

References:

Gao, H., Duan, Z., Cai, H. (2018) Understand the impacts of landscape features on the shape of storage capacity curve and its influence on flood, Hydrology Research, 49(1):

90-106.

Ye A, Duan Q, Yuan X, Wood EF, Schaake J. 2014. Hydrologic post-processing of MOPEX streamflow simulations. Journal of Hydrology 508: 147–156 DOI: 10.1016/j.jhydrol.2013.10.055

Kollat, J. B., P. M. Reed, and T. Wagener. "When are multiobjective calibration trade‐offs in hydrologic models meaningful?." Water Resources Research 48.3(2012):3520.

Rempe, D. M., and W. E. Dietrich (2014), A bottom-up control on fresh-bedrock topography under landscapes, Proc. Natl. Acad. Sci. U. S. A., 111(18), 6576–6581, doi:10.1073/pnas.1404763111.

Gomes GJC, Vrugt JA, Vargas EA. 2016. Toward improved prediction of the bedrock depth underneath hillslopes: Bayesian inference of the bottom-up control hypothesis using high-resolution topographic data. Water Resources Research 52 (4): 3085–3112

---

## Author Response (AR1)

**Replies to Anonymous Referee #1**

**This paper proposes a new method to estimate the spatial distribution of soil water storage capacity in catchments based on DEM and long-term water balance data. The shape parameter of the distribution is estimated based on HAND values derived from DEM. The average storage capacity over a catchment is estimated, by the MCT method, from climatological and vegetation data (i.e., long-term precipitation, runoff and seasonal NDVI). This method is evaluated in an experimental catchment and MOPEX catchments. This paper provides a novel method to estimate the spatial distribution of soil water storage capacity, which is an important feature for catchment hydrology but is usually not available. I have some comments for the authors to consider before the publication.**

Reply: We thank the Anonymous Reviewer #1 for recognising the innovation and the importance of this paper. We also appreciate all his/her constructive comments, which are valuable to improve the quality of this manuscript. For the detailed comments, please find our responses in below.

**Comments:**

1. **As shown in Figure 2, saturated area and runoff coefficient is dependent on the initial soil moisture condition ($S_u/S_{uMax}$). However, within a time step (daily in this paper), rainfall in the early time causes the increases of $S_u/S_{uMax}$ and runoff coefficient. Therefore, the runoff coefficient during a day is also affected by rainfall, and runoff coefficient is a function of $S_u/S_{uMax}$, $\beta$, and $S_{uMax}/P_e$ (Moore, 1985; Wang, 2018). Given the values of $S_u/S_{uMax}$ and $\beta$, equation 12 in Table 2 underestimates $R_u/P_e$ in a day, and this underestimation increases with increasing rainfall depth. Similarly for HSC module, if $A_s$ is determined by $S_u/S_{uMax}$ at the beginning of a day, the effect of $P_e$ on runoff coefficient is not considered.**

   Reply: The influence of $S_{uMax}/P_e$ on runoff coefficient estimation (Moore, 1985; Wang, 2018) has been discussed in the second paragraph of Section 6.3.

2. **For HSC and HSC-MCT modules: the relationship $As=f(S_u/S_{uMax})$ is obtained from the HAND values (Figure 2d). This relationship can be obtained by fitting a distribution function to the CDF of normalized HAND values (e.g., Figure 2b). For example, if the distribution used in Xinanjiang (Zhao, 1992) and VIC (Wood et al., 1992) is applied, the CDF of storage capacity is:**

$$F(C) = 1 - \left(1 - \frac{C}{C_m}\right)^{\beta}$$

   **The mean value of storage capacity is:**

$$S_{uMax} = \frac{C_m}{\beta + 1}$$

   **Substituting $C_m$ from equation (2) into equation (1), we obtain**

$$F(C) = 1 - \left(1 - \frac{1}{\beta + 1}\frac{C}{S_{uMax}}\right)^{\beta}$$

**c/$S_{uMax}$ is the normalized HAND values (HAND values divided by its average) in Figure 2b. The value of $\beta$ is estimated by fitting the distribution. Then the relationship $A_s = f(S_u/S_{uMax})$ is obtained:**

$$A_S = 1 - \left(1 - \frac{S_u}{S_{uMax}}\right)^{\frac{\beta}{\beta+1}}$$

**The curve for $A_s \sim S_u/S_{uMax}$ is concave since $\beta/(\beta+1) < 1$. $A_s \sim S_u/S_{uMax}$ for HBV is concave or convex. The distribution corresponding to the SCS curve number method is another alternative (Wang, 2018), and the CDF is written as:**

$$F(C) = 1 - \frac{1}{a} + \frac{\frac{C}{S_{uMax}} + 1 - a}{a\sqrt{\left(1 + \frac{C}{S_{uMax}}\right)^2 - 2a\frac{C}{S_{uMax}}}}$$

**Where $a$ is the shape parameter. The $A_s \sim S_u/S_{uMax}$ relation for equation (5) is:**

$$A_S = 1 - \frac{1}{a} + \frac{\frac{C}{S_{uMax}} + 1 - a}{a\sqrt{\left(1 + \frac{C}{S_{uMax}}\right)^2 - 2a\frac{C}{S_{uMax}}}}$$

**Where** $\dfrac{C}{S_{uMax}} = \dfrac{\left(2 - a\frac{S_u}{S_{uMax}}\right)\frac{S_u}{S_{uMax}}}{2\left(1 - \frac{S_u}{S_{uMax}}\right)}$ **It is interesting to test the goodness-of-fit of these two distributions (equations 3 and 5) to the empirical $A_s \sim S_u/S_{uMax}$ from the HAND-based values.**

Reply: In the manuscript, we have compared the model performance of HSC and HSC-MCT with HBV and TOPMODEL (as benchmarks), and found that the HSC module performed better in both calibration and validation. HBV is a good benchmark, because it has a relatively straightforward way of representing the runoff threshold in the root zone, albeit by calibration. TOPMODEL is also a good benchmark, because it uses a topographical index to define the runoff threshold. In our approach, the spatial distribution of the HAND values is used to derive the spatial distribution of the runoff (connectivity) thresholds, but from another topographical perspective than TOPMODEL. We agree that it would be interesting to test the goodness-of-fit of the Cumulative Distribution Function (CDF) of HSC with not only the HBV, but also the Xinanjiang, GR4J and SCS models. However it might be worthwhile to clarify that the intention of this study is to propose a new runoff generation module (HSC), which is, to some extent, supported by large-sample ecological field observation, and free of calibration, rather than comparing the CDF of HSC with other existing modules. More details can be found in the second paragraph of Section 6.3.

3. **Line 167: I guess that "SEF" represents "saturation excess flow". But spell out "SEF" which is not defined before. Same for SOF and SFF.**
   Reply: The full names of the SEF, and SOF have be clearly defined.

4. **It may be good to add a map to show the spatial distribution of $S_u$.**
   Reply: The $S_{uMax}$ for each MOPEX catchment in the HSC-MCT module was obtained in our previous study (Gao et al., 2014). We used the amount of root zone storage capacity, which ecosystems need to overcome drought periods (dry spells) with 20 years return period ($S_{R20y}$), as a proxy for $S_{uMax}$. The details of the method to derive the $S_{R20y}$ can be found in Gao et al., 2014.

5. **For figures with subplots, it is better to add "(a)" and "(b)", e.g., Figure 7 and Figure 8.**
   Reply: Done

6. **In the comparison of Figure 7a, is it possible to convert the distribution of TWI information to $A_s \sim S_u/S_{uMax}$ and add it to Figure 7a?**
   Reply: It is a good suggestion to put the TOPMODEL and HBV curves together, and compare their shape. But it is a difficult task, due to the different model assumption and concept. And to our best knowledge, we haven't found similar studies that systematically compare the TOPMODEL curves with HBV curves, which might indicate that this is not an easy task to be performed within a short time. Furthermore, in this study we just wanted to compare the model performance of HSC with HBV and TOPMODEL, rather than to unify all model approaches (see Section 6.3).

7. **In Figure S4 (catchment 08171000), the calibrated $\beta$ for HBV is about 1.8. $\beta$ =1.35 is very close to the curve of HSC. But it seems that the performance of HSC is better than HBV (lines 500-501 and Figure S3). Since $\beta$ of HBV is calibrated, why is the calibrated value of $\beta$ not around 1.35? Is this due to the effect of other calibrated parameters? Discussion on this may be helpful.**
   Reply: The effect of other calibrated parameters on model calibration and efficiency has be discussed in the third paragraph of Section 6.3.

8. **This comment is related to the previous one. For the comparison between HSC and HSC-MCT, is the calibrated $S_{uMax}$ for HSC similar to the estimated $S_{uMax}$ by MCT method? How about the other calibrated parameters (e.g., $D$, $K_f$, $K_s$, $T_{lagF}$) among the models (HBV, TOPMODEL HSC, and HSC-MCT)?**
   Reply: The comparison of the calibrated $S_{uMax}$ and the estimated $S_{uMax}$ by MCT can be found in Gao et al., 2014. For the other calibrated parameters, their effect on model performance will be discussed in the revised manuscript. It is worth noting that all models use the same model structure and prior range of remaining parameters (i.e. interception and response modules) to exclude the impact of other processes, and guarantee that the comparison of runoff generation modules is fair (see Section 2.4 and the third paragraph of Section 6.3).

**References:**

Moore, R. J. (1985), The probability-distributed principle and runoff production at point and basin scales, Hydrol. Sci. J., 30, 273-297.

Wang, D.: A new probability density function for spatial distribution of soil water storage capacity leads to SCS curve number method, Hydrol. Earth Syst. Sci. Discuss., https://doi.org/10.5194/hess-2018-32, in review, 2018.

Gao H, Hrachowitz M, Schymanski SJ, Fenicia F, Sriwongsitanon N, Savenije HHG. 2014. Climate controls how ecosystems size the root zone storage capacity at catchment scale. Geophysical Research Letters 41 (22): 7916–7923 DOI: 10.1002/2014gl061668

**Replies to Anonymous Referee #2**

**This study proposed the HAND-based storage capacity curve (HSC) for runoff generation parameterization in hydrological models. I like the idea to provide parameter reduced modules for hydrological modeling, considering the significant uncertainty in parameter calibration. However, the benefits for reducing parameter uncertainty by the HSC module were not illustrated by the current results. The authors claimed that the HSC/HSC-MCT module possess higher robustness and bear the potential to be implemented in prediction of ungauged basins, which however are not convincing from the current results. The benefits of the HSC/HSC-MCT modules need to be further discussed.**

Reply: We thank the Anonymous Referee #2 for the very constructive and detailed comments. For the benefits of HSC/HSC-MCT modules, we have added the results of model validation in the BB catchment (first paragraph of Section 4.2), and the model performance evaluated by $I_{KGL}$ in the MOPEX catchments (first paragraph of Section 5.2). Moreover, the limitations of the two modules are also discussed in Section 6.3. Please find our point-by-point responses to your comments in below.

**First, the HSC module obviously overestimated the saturated area fraction in the Bruntland Burn (BB) basin. The improvement for the correlation from 0.5 (TOPMODEL) to 0.6 (HSC) is rather small, which does not make any sense for illustrating the reduced deviations between the observed and simulated saturated area fractions. The model performance in validation period were not evaluated in the BB basin. Although the HSC module gained performance improvements on high flows in both calibration and validation periods in many MOPEX basins, the performance gains on low flows were not investigated. Moreover, the non-parameter HSC-MCT module produced much lower performance than the HSC module in many MOPEX cases. The gains for the HSC module should be attributed to the parameter calibration procedure, and potentially demonstrated the failure of the MCT module for many MOPEX cases.**

Reply:

*Overestimation of the saturated area.* The overestimation of the saturated area is most likely caused by the different definitions of saturated areas used in field measurements (saturated soils connected to the stream network as detected by the "squishy boots method") and in hydrological models (areas with potential for water accumulation across the catchment). The discussion and interpretation of the overestimation of the saturated area fraction in the BB basin are described in the third paragraph of Section 6.3. And we have further clarified the method to obtain experimental data in the second paragraph of Section 3.1.

*Model validation in BB basin.* We now have added the model performance for a validation period (2009-2014) and evaluated the models in the BB (see the first paragraph of Section 4.2).

*Model performance on low flow.* The performance gains on low flow ($I_{KGL}$) have been investigated and are shown in the supplementary figure S3. The results of $I_{KGL}$ illustrate that our proposed modules (HSC and HSC-MCT) also performed better in low flow simulation.

**Second, it is not fair to call the proposed modules are calibration-free. The HSC module also implied two parameters for the model application. The stream initiation threshold area was not included for**

the model calibration, but was tested to calculate the HAND values. I guess this threshold area should be tested in calculation experiments to prepare the model results using the HSC modules. The effects of this parameter on the performance of the HSC modules were not investigated in the results. Including this parameter in the calibration procedure would most likely to improve the model performance. Moreover, model calibration procedures are required to determine the remaining parameter values in Table 1 for both the applications of HSC and HSC-MCT modules. The benefits to reduce parameter uncertainty by excluding one-two parameters by the HSC and HSC-MCT modules were not clear. Considering the preparation of HAND values using DEM dataset in the HSC and HSC-MCT modules, the computational cost should be much higher than the calibrated modules in HBV and TOPMODEL.

Reply:

*Calibration-free.* We may politely insist that the HSC-MCT is a calibration-free runoff generation module. We agree that the threshold area for stream initiation is important while generating HAND maps. But the threshold area can be determined based on observation rather than calibration, although the threshold area varies in different climate, geology and landscape classes. The limitation of the fixed threshold area has been discussed in the third paragraph of Section 6.3.

*MCT method.* MCT is an approach to estimate the $S_{uMax}$ by measurable input. But since we fixed this parameter as $S_{R20y}$ (the amount of root zone storage capacity, which ecosystems need to bridge droughts with 20 years return period), which may also vary in different ecosystems. Improving the MCT to allow more flexible estimation for different ecosystems will be promising to improve model performance, which is discussed in the first paragraph of Section 6.3.

*Computational cost.* The discussion on the computational cost has been added in the revised manuscript (see the third paragraph of Section 6.3).

**Some other major concerns on the results are listed as follow.**

**1. Figure 6, why not show the saturated area fraction simulated by the HBV module? Is the HBV module spatially discretized for the model application?**

Reply: The saturated area fraction simulated by HBV is presented in Figure 8b. But the HBV cannot explicitly generate the spatial discretization of saturation areas.

**2. Figure 7, why was the beta value of 0.98 used for the HBV module? Was this beta value derived from the model calibration? Have you tried other beta values for the comparison between HBV and HSC modules? Why not show this curve for the TOPMODEL module? What is your purpose to show the frequency of TWI? Could you also show the frequency of HAND?**

Reply: Yes, the beta value of 0.98 is the averaged calibrated value of beta. Please note that the intention of the HSC module is to propose a new runoff generation module, which is, to some extent, supported by large-sample ecological field observation, and free of calibration, rather than fitting the CDF of HSC with other existing curves/modules (see the second paragraph of Section 6.3). The purpose to show the TWI frequency of TOPMODEL is to demonstrate the curve that we used to estimate runoff in

TOPMODEL. The HSC curve in Figure 7 is derived from the spatial distribution of HAND, therefore the HAND distribution curve is not shown in this figure.

**3. Figure 8, the label for soil moisture was missed. It is very difficult to find the observed soil moisture (or you don't have?). Can you label (a-b) for the two subplots? For the second and fourth events, the TOPMODEL matched the observed saturated area fractions very well. How to explain this? Please refine the caption, which is very difficult to understand.**

Reply: We don't have the observed soil moisture data, but we have the data of observed saturated area proportion. Label a-b have been added in Figure 8, and the caption of Figure 8 has also been refined. TOPMODEL does perform better in the second and the fourth events, but generally HSC performs better than TOPMODEL (evaluated by $R^2$ and $I_{KGE}$) (Section 4.3).

**4. Figure 9, as I suggested before, a correlation coefficient does not make any sense to illustrate the deviations between the observed and simulated saturated area fractions, given that only seven observation events. Could you use some other metrics to compare the bias or deviation errors between the observed and simulated saturated area fractions?**

Reply: Thank you for the suggestion to use a different criteria to evaluate model performance to reproduce saturation areas. $I_{KGE}$ might be a better metric to evaluate model performance on saturated area fraction estimation. Evaluated by $I_{KGE}$, HSC also performs better than the TOPMODEL, although both HSC and TOPMODEL do not perform well (-3.0 for HSC, and -3.4 for TOPMODEL). The reasons for the unsatisfactory results are discussed in the third paragraph of Section 6.3.

**5. Figures 10-11, what do you intend to say from these figures?**

Reply: We intended to present the procedures to derive the HSC curves for the MOPEX catchments, which we believe are helpful for readers to understand how the HSC module works.

**6. Figure 12, I suggest to compare the values for IKGL as well. The models were calibrated on both IKGL and IKGE, there should be strong trade-off between these two objective functions. That means the HSC module possibly sacrificed the performance for low flows (IKGL) to improve the performance for high flows (IKGE). The evaluated modules mainly differed on the calculation of soil storage capacity, which has significant effect on the generation of low flows. In my opinion, performance for low flows should also be an important indicator for the validity of the runoff generation assumptions.**

Reply: The results of $I_{KGL}$ have been incorporated into the revised manuscript (Figure S3, and the first paragraph of Section 5.2).

**Minor concerns:**

**1. I suggest to remove "Calibration-free" from the title. HSC module needs to be calibrated, and the HSC-MCT module performed poorly in many MOPEX cases.**

Reply: We may politely insist that the HSC-MCT module is calibration-free and performs equally well or better as a calibrated model. There are two reasons. Firstly, as we clarified in the above, HSC is directly derived from the HAND distribution in a DEM, without any calibration. Secondly, HSC-MCT performs comparably well with HBV. Since the median $I_{KGE}$ value of HSC-MCT is 0.65, which is a better performance compared to HBV (0.61). And the averaged $I_{KGE}$ value of HSC-MCT is 0.59, which is comparable to 0.62 (HBV). The models' performance on $I_{KGL}$ have similar results. So, it is fair to say the model performance of the calibration-free HSC-MCT and HBV are comparable.

**2. Lines 30-33, I am not convinced to agree with this from the current results. What do you mean "facilitated effective visualization of the saturated area"? Is it important?**

Reply: This sentence has been rephrased.

**3. Introduction is too long from my taste. It is very difficult to get the motivations of this study from this section. I would suggest to refine it.**

Reply: We have tremendously refined and shortened the introduction.

**4. Lines 213-214, remove "Hydrological: : : inevitable". Line 217, what do you mean "HAND contours are parallel in runoff generation"? Is that possible derived from the DEM?**

Reply: This have been rephrased.

**5. Line 227, could you please add more details for the calculation of HAND values?**

Reply: This has been rephrased. And more details about the calculation of HAND can be found in Rennó et al., 2008; Gharari et al., 2011.

**6. Line 314, how did you define the pareto-frontier? Did you use the Euclidean distance or threshold values?**

Reply: The Pareto frontier is defined by Euclidean distance. Please refer to Vrugt et al., 2003.

**7. Section 3.1, could you please add some details on the climatic and hydrological data in the BB basin? Any ground gauged stations do you have there?**

Reply: Yes, we have added more but brief details on the climatic and hydrological data in the BB catchment.

**8. Also in section 3.1, could you introduce the spatial interpolation of the field mapping of the saturated areas?**

Reply: The saturation maps are not interpolated. They are generated directly by field mapping, and global positioning system (GPS) was used to delineate the boundary of saturation areas using the "squishy boot method" (Ali et al., 2014; Birkel et al., 2010). We have clarified this in the methods of the revised manuscript.

**9. Line 369-374, move to the methodology section.**

Reply: We have revised as suggested.

**10. Lines 415-420, move to the methodology section.**

Reply: We have revised as suggested.

**11. Lines 455-456, 'dramatically improved' may be not fair. 'simultaneously maintaining model robustness and consistency' is also not convinced by the results.**

Reply: We have revised as suggested.

**12. Lines 491-496, it is not fair to only discuss the cases where HSC/HSC-MCT outperformed the benchmark modules. Why not discuss the reasons for the cases where HSC/HSC-MCT produced lower performance?**

Reply: The results and the reasons for the cases where HSC/HSC-MCT produced lower performance have be described in the end of third and four paragraphs of Section 5.2, and the end of the first paragraph of Section 6.1.

**13. Discussion is also too much. It is difficult to get the main messages from the long text. Maybe remove lines 508-522, and lines 539-556.**

Reply: We have revised as suggested.

**14. Lines 646-647, maybe it is not so important to say as one of the conclusions here.**

Reply: We have revised as suggested.

**15. There are many sentence started with 'And', this is very strange (kind of grammatical error).**

Reply: We have revised the content thoroughly as suggested.

Reply: A list of the used catchments has been added as SI material.

**References:**

[revised manuscript text omitted]

---

## Author Response (AR2)

Dear Editor,

Thank you very much for your time and efforts regarding our manuscript. We highly appreciate the constructive comments from three reviewers that offer us the opportunity to clarify some concerns and further improve our manuscript.

Please find enclosed our detailed point-by-point responses to all the comments, as well as the uploaded revised version of our manuscript. The comments are black, our response in blue. For easy review, we have also used the "Track Changes" function in the revised manuscript to make our revisions more easily visible. The modifications are mainly threefold:

1. The motivation was emphasized by adding Figure 1 taken from Fan et al., PNAS 2017, showing the increase of rooting depth with the increase of HAND in most parts of hillslope. The HSC module provides a rational from an ecological perspective to understand the linkage between large-sample hillslope ecological observations and the curve of root zone storage capacity distribution (Figure 1, 2, 3)
2. The impact of catchment characteristics on HSC model performance was analyzed, including topography (averaged elevation, averaged HAND, averaged slope), geology (averaged depth to rock), soil texture (K factor), land use (forest cover proportion), and stream density. It was found that HSC performs better in the catchments with gentle topography, less forest cover and arid climate.
3. The discussion was improved, on the model comparison between HSC and TOPMODEL in the Bruntland Burn catchment.

We hope responses and revisions will satisfy all reviewers' comments. Thanks again and we are looking forward to receiving your decision.

Yours sincerely,

Hongkai Gao on behalf of all the co-authors

**Anonymous Referee #1**

accepted as is

**Anonymous Referee #2**

I am appreciated that the authors add more explanations and discussions to improve the manuscript. However, the benefits from the application of the new runoff generation module are still not clear to me. The authors discussed the model uncertainty in the introduction section, while the results have not addressed this issue in their manuscript. The ability of the proposed module to improve the conceptualization of real catchment behaviors in the hydrological model is not convincing. Some of other concerns are listed as follow.

Reply: We thank the Anonymous Referee #2 for his/her further comments and suggestions on the manuscript. Our detailed replies can be found below.

1.      The authors defined 'calibration free' as one of the major sell points of their work. However, the benefits from the reduced calibration work need to be further interpreted in the results. After the integration of the proposed runoff generation module, the hydrological model still needs calibration. The computation cost caused by the preparation of the HAND curves is even higher than the computation cost reduced by the smaller parameter space. For the practical application of hydrological modeling, running automatic calibration without any pre-calculation is more preferable, unless the authors can provide poofs for that the new runoff generation module is more close to the realism in the catchment. Figures 6-9 compared the simulated and observed saturated areas by various modules. However, I would say the HSC module has not shown higher performance than the Topmodel in Figures 8-9. The Topmodel shows higher performance at low saturated areas (Fig. 9a). Figure 6 has not evaluated the distribution of saturated areas produced by the HBV model, which can be also set up at grid scale. Figure 7 does not make any sense, considering observation is missing.

**Motivation and rational of HSC**

For the motivation of this study, we added a figure (Figure 1), taken from Fan et al., 2017 (with permission from PNAS), showing the increase of rooting depth with the increase of HAND in most parts of hillslope, only except for the very high HAND hillslopes. Figure 1 is the result of thousands of ecological measurements at global scale, which illustrates that the assumption of HSC likely fits well with catchment realism supported by a large dataset of field observations. The HSC module provides a rational from an ecological perspective to understand the linkage and mechanism between large-sample hillslope ecological observations and the curve of root zone storage capacity distribution (Figure 1, 2, 3).

The benefits of the new HSC module are two- fold. From a technical point of view, the HSC allows us to make Prediction in Ungauged Basins without calibrating the beta parameter in many conceptual hydrological models (e.g. HBV, Xinanjiang etc). But as the reviewer pointed out (which we also recognized) there are other modules with relatively parsimonious parameterizations (e.g. TOPMODEL

and GR4J), that can work well in terms of model performance. In contrast, the HSC module,
furthermore, from a scientific point of view, provides us with a new perspective on the linkage between
root zone storage capacity in both hillslopes and at catchment scales (long-term ecosystem evolution)
with insights into runoff generation (event scale rainfall-runoff generation).

Further asking questions of "why" rather than "what" likely leads to more useful insights and a new way
forward (McDonnell et al., 2007). Catchments are geomorphological and even ecological systems whose
parts are inter-related due to catchment self-organization and co-evolution (Sivapalan and Blöschl,
2015; Savenije and Hrachowitz, 2017).

**Computational cost**

The computational cost of the HSC is more expensive than HBV, and similar to TOPMODEL, due to the
cost of preprocessed topographic analysis. But once the Su-As curve is completed, the computational
cost is quite comparable with HBV.

**Interpretation of Figure 6**

For Figure 6 (now Figure 7 in the revised manuscript), we may politely disagree with the Anonymous
Reviewer 2. The HBV cannot generate the distribution of saturated areas at catchment scale. Since HBV
only calculates the runoff coefficient of certain rainfall events in a lumped way. It cannot map out the
saturated area variation. We think it is still worthwhile to compare the contributing area simulated by
the new HSC module and the TOPMODEL, since TOPMODEL is a benchmark in this study. Also, it might
be interesting to show the simulated contributing area compared to the observed saturated area,
despite that they are not exactly the same. In theory, the observed saturated area should be within the
simulated contributing area, due to the fact that the saturated soil moisture is always larger than field
capacity. From this point of view, we show that the observed saturated area is almost always within the
contributing area simulated by HSC, but TOPMODEL missed this important feature (July 2 and
September 2 in 2008) supporting our statement that HSC performed better in reproducing saturated
area variation.

We rephrased the discussion on model comparison between HSC and TOPMODEL. More details can be
found in Line 427-430, 666-668 in the revised manuscript (clear version hereinafter).

2.        The comparable or better performance produced by the model coupled with the proposed
runoff generation module may be caused by either the application of the new runoff generation module
or the calibration run with less parameters. The automatic calibration algorithm could produce higher
performance when the calibration parameter space is reduced. This need to be further analyzed.

**New module or less parameter?**

This is indeed a very good point. Parsimonious models (e.g. the GR4J, ref. Perrin et al., 2003), with
empirical curves similar to the HSC, likely result in good model performance. Parameter identifiability in
calibration is one of the reasons. However, the rationale of most models is still largely unknown, and
lack of the physically explanation to interpret these empirical curves described by mathematical
functions (e.g. Equation 3 in Perrin et al., 2003). (Line 576-580).

3.  The motivation of this sturdy is not clear enough. With the proposed module, more calculation work is needed and produced the same or bit better model performance. The model uncertainty has not been analyzed, and the transferability of the runoff generation module has not been investigated in this study.

In general, limitations in the current work certainly prevented more scientific contributions from this study. The authors should pay more efforts to make their motivation and results more convincing.

**Motivation**

The motivation can be found in our reply to the first comment and with all due respect the Reviewer does seem to miss the main point, which we tried to further emphasize in the revised manuscript: Since the HSC is an a priori module, which we do not calibrate, the module can be perfectly "transferred" to other catchments without calibration. Hence, the large sample application!

**Anonymous Referee #3**

I would like to thank the authors for their revision, particularly improving discussion section. For some of my comments, however, I do not see a clear response:

1)  I'm still not convinced by the use of BB basin to support the conclusion that the new model (HSC) outperforms the TOPMODEL ("We found that the HSC performed better in reproducing the spatio-temporal pattern of the observed saturated areas in the BB compared to TOPMODEL"). The Figures 6 and 8 clearly indicate that both models are significantly overestimating what is considered as ground truth here (observed pattern of saturated area). I do not understand well the argument that the values are not directly comparable. If so, why to compare them? This part is not well linked with the validation of the new approach in its current form.

Thank you for your comment, which we tried to clarify in our revised manuscript. We have rephrased the statements in Section 4.3 and Line 640-644.

**Section 4.3**:

Comparing the estimated contributing area of TOPMODEL with the HSC module, we found the results of the HSC correlates better (R2=0.60, IKGE=-3.0) with the observed saturated areas than TOPMODEL (R2=0.50, IKGE=-3.4) (Figure 10). For spatial patterns, the HSC contributing area is located close to the river network, and reflects the spatial pattern of observed saturated area. While TOPMODEL results are more scattered, probably due to the sensitivity of TWI to DEM resolution (Figure 7). The HSC is more discriminating in terms of less frequently giving an unrealistic 100% catchment saturation retaining parts of the unsaturated upper hillslopes.

**Line 640-644:**

Interestingly, in theory the observed saturated area should be within the simulated contributing area, due to the fact that the saturated soil moisture is always larger than field capacity. From this point of view, the observed saturated area is smaller and within the contributing area simulated by HSC, but
TOPMODEL missed this important feature.

2)      In some places of section 5, it will be interesting to see more specific interpretation of the
results. E.g. for statement "Figure 11c interestingly shows that in some catchments, there is almost no
threshold…" it will interesting to see some more specific generalisation for which catchments it applies.
Or "…where the HSC model performed better are mostly located in the Great Plains, with modest
sloping (4.0 degree)…" Does it apply/relate to all catchments with slope 4degree or even less ("HSC
outperformed catchments have flat terrain (2.3 degree) with moderate averaged HAND value (26m)"?
Some more evaluations (or more specific formulations) allowing some more specific generalisation of
results will be interesting here. Where one can expect the new model is better/worser than
TOPMODEL/HBV in other regions – outside US?

This is an excellent question, thank you, which helped us to improve the presentation of this work. We
did a systematic analysis between model performance and catchment characteristics included in the
revised manuscript. See Line 494-505, and the new Table 3 and Table 4.

3)      Figure 6 legend is still confusing for me. Would it be possible to make all three
columns/methods of maps just with binary colours (is/is not saturated)?

Thank you. This was changed in the revised manuscript.

4)      Please add the new references to the list.

Done.

[revised manuscript text omitted]
 \ (11)$$ | $$\frac{R_u}{P_e} = \left(\frac{S_u}{S_{uMax}}\right)^{\beta} \ (12)^{*}$$ |

$$\frac{E_a}{E_p - E_i} = \frac{S_u}{C_e S_{uMax}} \quad (13)$$

| Splitter and Lag function | | $R_f = R_u D \ (17); \ R_s = R_u (1-D) \ (14)$ |
| --- | --- | --- |

$$R_{fl}(t) = \sum_{i=1}^{T_{lagf}} c_f(i) \cdot R_f(t-i+1) \quad (15)$$

$$c_f(i) = i / \sum_{u=1}^{T_{lagf}} u \quad (16)$$

| Fast reservoir | $\dfrac{dS_f}{dt} = R_f - Q_f \ (17)$ | $Q_f = S_f / K_f \ (18)$ |
| --- | --- | --- |
| Slow reservoir | $\dfrac{dS_s}{dt} = R_s - Q_s \ (19)$ | $Q_s = S_s / K_s \ (20)$ |

Table 3. Data source of the MOPEX catchments.

[revised manuscript text omitted]

and TOPMODEL, for the 323 MOPEX catchments.

---

## Author Response (AR3)

This paper presents a very interesting work to develop a simple topography-driven and calibration-free
runoff generation module. The module works for saturation excess runoff generation mechanism, which
prevails in most humid/semi-humid areas and is demonstrated by some experts to operate in some arid
areas also. The module was rigorously compared against the corresponding models in HBV and
TOPMODEL. The experiments in both data-rich experimental watersheds and MOPEX catchments
support the superiority of the new module (called HSC and HSC-MCT). The authors also discuss the deep
reason of why type question (why can HSC outperforms calibrated-type module) in the context of
ecological evolution theory. The proposed method has a wide implication for hydrological and ecological
research.

We thank the Editor's positive comments.

Some minor comments are listed below for authors' reference:
1. P5L125, one or two sentences should add to explain MCT concisely. This term is not a popular one in
hydrological literature. No further explanation will hinder the reader's understanding.

More explanation of MCT is added. (L158-161)

2. P8L214, the term of subsurface flow. Quite a few different terms have been used for the flow in soil
media. The authors can refer to Markus Weiler and Jeffery MacDonnell (in Encyclopedia of Hydrological
Sciences. Edited by M G Anderson.). In my mind, the term subsurface flow could refer to all kinds of flow
types occurring in soil media, including soil matrix flow, preferential flow, or others. I understand the
authors mean preferential type flow by subsurface flow here.

This sentence is modified (L247-248). And we add more interpretation in the discussion. (L585-590)

3. P16L440, 'interestingly' is not suitable here, because HAND by its definition should not depend on
elevation.
Changed. (L477-479)

[revised manuscript text omitted]